# Exponentially vanishing sub-optimal local minima in multilayer neural networks

## Abstract

**Background:** Statistical mechanics results (Dauphin et al. (2014); Choromanska et al. (2015)) suggest that local minima with high error are exponentially rare in high dimensions. However, to prove low error guarantees for Multilayer Neural Networks (MNNs), previous works so far required either a heavily modified MNN model or training method, strong assumptions on the labels (*e.g.*, "near" linear separability), or an unrealistically wide hidden layer with $\Omega(N)$ units.

**Results:** We examine a MNN with one hidden layer of piecewise linear units, a single output, and a quadratic loss. We prove that, with high probability in the limit of $N \to \infty$ datapoints, the volume of differentiable regions of the empiric loss containing sub-optimal differentiable local minima is exponentially vanishing in comparison with the same volume of global minima, given standard normal input of dimension $d_0 = \tilde{\Omega}\left(\sqrt{N}\right)$, and a more realistic number of $d_1 = \tilde{\Omega}(N/d_0)$ hidden units. We demonstrate our results numerically: for example, $0\%$ binary classification training error on CIFAR with only $N/d_0 \approx 16$ hidden neurons.

## 1 Introduction

**Motivation.** Multilayer Neural Networks (MNNs), trained with simple variants of stochastic gradient descent (SGD), have achieved state-of-the-art performances in many areas of machine learning (LeCun et al., 2015). However, theoretical explanations seem to lag far behind this empirical success (though many hardness results exist, *e.g.*, (Síma, 2002; Shamir, 2016)). For example, as a common rule-of-the-thumb, a MNN should have at least as many parameters as training samples. However, it is unclear why such over-parameterized MNNs often exhibit remarkably small generalization error (*i.e.*, difference between "training error" and "test error"), even without explicit regularization (Zhang et al., 2017a).

Moreover, it has long been a mystery why MNNs often achieve low training error (Dauphin et al., 2014). SGD is only guaranteed to converge to critical points in which the gradient of the expected loss is zero (Bottou, 1998), and, specifically, to local minima (Pemantle, 1990) (this is true also for regular gradient descent (Lee et al., 2016)). Since loss functions parameterized by MNN weights are non-convex, it is unclear why does SGD often work well – rather than converging to sub-optimal local minima with high training error, which are known to exist (Fukumizu & Amari, 2000; Swirszcz et al., 2016). Understanding this behavior is especially relevant in important cases where SGD does get stuck (He et al., 2016) – where training error may be a bottleneck in further improving performance.

Ideally, we would like to quantify the probability to converge to a local minimum as a function of the error at this minimum, where the probability is taken with the respect to the randomness of the initialization of the weights, the data and SGD. Specifically, we would like to know, under which conditions this probability is very small if the error is high, as was observed empirically (*e.g.*, (Dauphin et al., 2014; Goodfellow et al., 2015)). However, this seems to be a daunting task for realistic MNNs, since it requires a characterization of the sizes and distributions of the basins of attraction for all local minima.

Previous works (Dauphin et al., 2014; Choromanska et al., 2015), based on statistical physics analogies, suggested a simpler property of MNNs: that with high probability, local minima with high error diminish exponentially with the number of parameters. Though proving such a geometric property with realistic assumptions would not guarantee convergence to global minima, it appears to

be a necessary first step in this direction (see discussion on section 6). It was therefore pointed out as an open problem at the Conference of Learning Theory (COLT) 2015. However, one has to be careful and use realistic MMN architectures, or this problem becomes "too easy".

For example, one can easily achieve zero training error (Nilsson, 1965; Baum, 1988) – if the MNN's last hidden layer has more neurons than training samples. Such extremely wide MNNs are easy to optimize (Yu, 1992; Huang et al., 2006; Livni et al., 2014; Haeffele & Vidal, 2015; Shen, 2016; Nguyen & Hein, 2017). In this case, the hidden layer becomes linearly separable in classification tasks, with high probability over the random initialization of the weights. Thus, by training the last layer we get to a global minimum (zero training error). However, such extremely wide layers are not very useful, since they result in a huge number of weights, and serious overfitting issues. Also, training only the last layer seems to take little advantage of the inherently non-linear nature of MNNs.

Therefore, in this paper we are interested to understand the properties of local and global minima, but at a more practical number of parameters – and when at least two weight layers are trained. For example, Alexnet (Krizhevsky, 2014) is trained using about 1.2 million ImageNet examples, and has about 60 million parameters – 16 million of these in the two last weight layers. Suppose we now train the last two weight layers in such an over-parameterized MNN. When do the sub-optimal local minima become exponentially rare in comparison to the global minima?

**Main contributions.** We focus on MNNs with a single hidden layer and piecewise linear units, optimized using the Mean Square Error (MSE) in a supervised binary classification task (Section 2). We define $N$ as the number of training samples, $d_l$ as the width of the $l$-th activation layer, and $g(x) \dot{<} h(x)$ as an asymptotic inequality in the leading order (formally: $\lim_{x \to \infty} \frac{\log g(x)}{\log h(x)} < 1$). We examine Differentiable Local Minima (DLMs) of the MSE: sub-optimal DLMs where at least a fraction of $\epsilon > 0$ of the training samples are classified incorrectly, and global minima where all samples are classified correctly.

Our main result, Theorem 10, states that, with high probability, the total volume of the differentiable regions of the MSE containing sub-optimal DLMs is exponentially vanishing in comparison to the same volume of global minima, given that:

**Assumption 1.** *The datapoints (MNN inputs) are sampled from a standard normal distribution.*

**Assumption 2.** $N \to \infty$, $d_0(N)$ *and* $d_1(N)$ *increase with* $N$, *while* $\epsilon \in (0, 1)$ *is a constant*[1].

**Assumption 3.** *The input dimension scales as* $\sqrt{N} \dot{<} d_0 \dot{\leq} N$.

**Assumption 4.** *The hidden layer width scales as*

$$\frac{N \log^4 N}{d_0} \dot{<} d_1 \dot{<} N .\tag{1.1}$$

Importantly, we use a standard, unmodified, MNN model, and make no assumptions on the target function. Moreover, as the number of parameters in the MNN is approximately $d_0 d_1$, we require only "asymptotically mild" over-parameterization: $d_0 d_1 \dot{>} N \log^4 N$ from eq. (1.1). For example, if $d_0 \propto N$, we only require $d_1 \dot{>} \log^4 N$ neurons. This improves over previously known results (Yu, 1992; Huang et al., 2006; Livni et al., 2014; Shen, 2016; Nguyen & Hein, 2017) – which require an extremely wide hidden layer with $d_1 \geq N$ neurons (and thus $N d_0$ parameters) to remove sub-optimal local minima with high probability.

In section 5 we validate our results numerically. We show that indeed the training error becomes low when the number of parameters is close to $N$. For example, with binary classification on CIFAR and ImageNet, with only 16 and 105 hidden neurons (about $N/d_0$), respectively, we obtain less then $0.1\%$ training error. Additionally, we find that convergence to non-differentiable critical points does not appear to be very common.

Lastly, in section 6 we discuss our results might be extended, such as how to apply them to "mildly" non-differentiable critical points.

**Plausibility of assumptions.** Assumption 1 is common in this type of analysis (Andoni et al., 2014; Choromanska et al., 2015; Xie et al., 2016; Tian, 2017; Brutzkus & Globerson, 2017). At first it may

---

[1]For brevity we will usually keep implicit the $N$ dependencies of $d_0$ and $d_1$.

appear rather unrealistic, especially since the inputs are correlated in typical datasets. However, this no-correlation part of the assumption may seem more justified if we recall that datasets are many times whitened before being used as inputs. Alternatively, if, as in our motivating question, we consider the input to the our simple MNN to be the output of the previous layers of a deep MNN with fixed random weights, this also tends to de-correlate inputs (Poole et al., 2016, Figure 3). The remaining part of assumption 1, that the distribution is normal, is indeed strong, but might be relaxed in the future, *e.g.* using central limit theorem type arguments.

In assumption 2 we use this asymptotic limit to simplify our proofs and final results. Multiplicative constants and finite (yet large) $N$ results can be found by inspection of the proofs. We assume a constant error $\epsilon$ since typically the limit $\epsilon \to 0$ is avoided to prevent overfitting.

In assumption 3, for simplicity we have $d_0 \dot{\leq} N$, since in the case $d_0 \geq N$ the input is generically linearly separable, and sub-optimal local minima are not a problem (Gori & Tesi, 1992; Safran & Shamir, 2016). Additionally, we have $\sqrt{N} \dot{<} d_0$, which seems very reasonable, since for example, $d_0/N \approx 0.016$, $0.061$ and $0.055$ MNIST, CIFAR and ImageNet, respectively.

In assumption 4, for simplicity we have $d_1 \dot{<} N$, since, as mentioned earlier, if $d_1 \geq N$ the hidden layer is linearly separable with high probability, which removes sub-optimal local minima. The other bound $N \log^4 N \dot{<} d_0 d_1$ is our main innovation – a large over-parameterization which is nevertheless asymptotically mild and improves previous results.

**Previous work.** So far, general low (training or test) error guarantees for MNNs could not be found – unless the underlying model (MNN) or learning method (SGD or its variants) have been significantly modified. For example, (Dauphin et al., 2014) made an analogy with high-dimensional random Gaussian functions, local minima with high error are exponentially rare in high dimensions; (Choromanska et al., 2015; Kawaguchi, 2016) replaced the units (activation functions) with independent random variables; (Pennington & Bahri, 2017) replaces the weights and error residuals with independent random variables; (Baldi, 1989; Saxe et al., 2014; Hardt & Ma, 2017; Lu & Kawaguchi, 2017; Zhou & Feng, 2017) used linear units; (Zhang et al., 2017b) used unconventional units (*e.g.*, polynomials) and very large hidden layers ($d_1 = \text{poly}(d_0)$, typically $\gg N$); (Brutzkus & Globerson, 2017; Du et al., 2017; Shalev-Shwartz et al., 2017) used a modified convnet model with less then $d_0$ parameters (therefore, not a universal approximator (Cybenko, 1989; Hornik, 1991)); (Tian, 2017; Soltanolkotabi et al., 2017; Li & Yuan, 2017) assume the weights are initialized very close to those of the teacher generating the labels; and (Janzamin et al., 2015; Zhong et al., 2017) use a non-standard tensor method during training. Such approaches fall short of explaining the widespread success of standard MNN models and training practices.

Other works placed strong assumptions on the target functions. For example, to prove convergence of the training error near the global minimum, (Gori & Tesi, 1992) assumed linearly separable datasets, while (Safran & Shamir, 2016) assumed strong clustering of the targets ("near" linear-separability). Also, (Andoni et al., 2014) showed a $p$-degree polynomial is learnable by a MNN, if the hidden layer is very large ($d_1 = \Omega\left(d_0^{6p}\right)$, typically $\gg N$) so learning the last weight layer is sufficient. However, these are not the typical regimes in which MNNs are required or used. In contrast, we make no assumption on the target function. Other closely related results (Soudry & Carmon, 2016; Xie et al., 2016) also used unrealistic assumptions, are discussed in section 6, in regards to the details of our main results.

Therefore, in contrast to previous works, the assumptions in this paper are applicable in *some* situations (*e.g.*, Gaussian input) where a MNN trained using SGD might be used and be useful (*e.g.*, have a lower test error then a linear classier).

## 2 PRELIMINARIES AND NOTATION

**Model.** We examine a Multilayer Neural Network (MNN) with a single hidden layer and a scalar output. The MNN is trained on a finite training set of $N$ datapoints (features) $\mathbf{X} \triangleq \left[\mathbf{x}^{(1)}, \ldots, \mathbf{x}^{(N)}\right] \in \mathbb{R}^{d_0 \times N}$ with their target labels $\mathbf{y} \triangleq \left[y^{(1)}, \ldots, y^{(N)}\right]^\top \in \{0,1\}^N$ – each datapoint-label pair $\left(\mathbf{x}^{(n)}, y^{(n)}\right)$ is independently sampled from some joint distribution $\mathbb{P}_{X,Y}$. We

define $\mathbf{W} = [\mathbf{w}_1, \dots, \mathbf{w}_{d_1}]^\top \in \mathbb{R}^{d_1 \times d_0}$ and $\mathbf{z} \in \mathbb{R}^{d_1}$ as the first and second weight layers (bias terms are ignored for simplicity), respectively, and $f(\cdot)$ as the common leaky rectifier linear unit (LReLU (Maas et al., 2013))

$$f(u) \triangleq ua(u) \ \text{ with } a(u) \triangleq \begin{cases} 1 & , \text{if }, u > 0 \\ \rho & , \text{if } u < 0 \end{cases}, \tag{2.1}$$

for some $\rho \neq 1$ (so the MNN is non-linear), where both functions $f$ and $a$ operate component-wise (*e.g.*, for any matrix $\mathbf{M}$: $(f(\mathbf{M}))_{ij} = f(M_{ij})$). Thus, the output of the MNN on the entire dataset can be written as

$$f(\mathbf{W}\mathbf{X})^\top \mathbf{z} \in \mathbb{R}^N. \tag{2.2}$$

We use the mean square error (MSE) loss for optimization

$$\text{MSE} \triangleq \frac{1}{N} \|\mathbf{e}\|^2 \ \text{ with } \mathbf{e} \triangleq \mathbf{y} - f(\mathbf{W}\mathbf{X})^\top \mathbf{z}, \tag{2.3}$$

where $\|\cdot\|$ is the standard euclidean norm. Also, we measure the empiric performance as the fraction of samples that are classified correctly using a decision threshold at $y = 0.5$, and denote this as the mean classification error, or MCE[2]. Note that the variables $\mathbf{e}$, MSE, MCE and other related variables (*e.g.*, their derivatives) all depend on $\mathbf{W}, \mathbf{z}, \mathbf{X}, \mathbf{y}$ and $\rho$, but we keep this dependency implicit, to avoid cumbersome notation.

**Additional Notation.** We define $g(x) \dot{<} h(x)$ if and only if $\lim_{x \to \infty} \frac{\log g(x)}{\log h(x)} < 1$ (and similarly $\dot{\leq}$ and $\dot{=}$). We denote "$\mathbf{M} \sim \mathcal{N}$" when $\mathbf{M}$ is a matrix with entries drawn independently from a standard normal distribution (*i.e.*, $\forall i, j\colon M_{ij} \sim \mathcal{N}(0,1)$). The Khatari-rao product (*cf.* (Allman et al., 2009)) of two matrices, $\mathbf{A} = [\boldsymbol{a}^{(1)}, \dots, \boldsymbol{a}^{(N)}] \in \mathbb{R}^{d_1 \times N}$ and $\mathbf{X} = [\mathbf{x}^{(1)}, \dots, \mathbf{x}^{(N)}] \in \mathbb{R}^{d_0 \times N}$ is defined as

$$\mathbf{A} \circ \mathbf{X} \triangleq [\boldsymbol{a}^{(1)} \otimes \mathbf{x}^{(1)}, \dots, \boldsymbol{a}^{(N)} \otimes \mathbf{x}^{(N)}] \in \mathbb{R}^{d_0 d_1 \times N}, \tag{2.4}$$

where $\boldsymbol{a} \otimes \mathbf{x} = [a_1 \mathbf{x}^\top, \dots, a_{d_1} \mathbf{x}^\top]^\top$ is the Kronecker product.

## 3 Basic Properties of Differentiable Local minima

MNNs are typically trained by minimizing the loss over the training set, using Stochastic Gradient Descent (SGD), or one of its variants (*e.g.*, Adam (Kingma & Ba, 2014)). Under rather mild conditions (Pemantle, 1990; Bottou, 1998), SGD asymptotically converges to local minima of the loss. For simplicity, we focus on differentiable local minima (DLMs) of the MSE (eq. (2.3)). In section 4 we will show that sub-optimal DLMs are exponentially rare in comparison to global minima. Non-differentiable critical points, in which some neural input (pre-activation) is exactly zero, are shown to be numerically rare in section 5, and are left for future work, as discussed in section 6.

Before we can provide our results, in this section we formalize a few necessary notions. For example, one has to define how to measure the amount of DLMs in the over-parameterized regime: there is an infinite number of such points, but they typically occupy only a measure zero volume in the weight space. Fortunately, using the differentiable regions of the MSE (definition 1), the DLMs can partitioned to a finite number of equivalence groups, so all DLMs in each region have the same error (Lemma 2). Therefore, we use the volume of these regions (definition 3) as the relevant measure in our theorems.

**Differentiable regions of the MSE.** The MSE is a piecewise differentiable function of $\mathbf{W}$, with at most $2^{d_1 N}$ differentiable regions, defined as follows.

**Definition 1.** For any $\mathbf{A} \in \{\rho, 1\}^{d_1 \times N}$ we define the corresponding differentiable region

$$\mathcal{D}_{\mathbf{A}}(\mathbf{X}) \triangleq \{\mathbf{W} | a(\mathbf{W}\mathbf{X}) = \mathbf{A}\} \subset \mathbb{R}^{d_1 \times d_0}. \tag{3.1}$$

Also, any DLM $(\mathbf{W}, \mathbf{z})$, for which $\mathbf{W} \in \mathcal{D}_{\mathbf{A}}(\mathbf{X})$ is denoted as "in $\mathcal{D}_{\mathbf{A}}(\mathbf{X})$".

---

[2]Formally (this expression is not needed later): $\text{MCE} \triangleq \frac{1}{2N} \sum_{n=1}^{N} \left[ 1 + \left(1 - 2y^{(n)}\right) \text{sign}\left(e^{(n)} - \frac{1}{2}\right) \right]$.

Note that $\mathcal{D}_{\mathbf{A}}(\mathbf{X})$ is an open set, since $a(0)$ is undefined (from eq. 2.1). Clearly, for all $\mathbf{W} \in \mathcal{D}_{\mathbf{A}}(\mathbf{X})$ the MSE is differentiable, so any local minimum can be non-differentiable only if it is not in any differentiable region. Also, all DLMs in a differentiable region are equivalent, as we prove on appendix section 7:

**Lemma 2.** *At all DLMs in $\mathcal{D}_{\mathbf{A}}(\mathbf{X})$ the residual error $\mathbf{e}$ is identical, and furthermore*

$$(\mathbf{A} \circ \mathbf{X})\,\mathbf{e} = 0\,. \tag{3.2}$$

The proof is directly derived from the first order necessary condition of DLMs ($\nabla \mathrm{MSE} = 0$) and their stability. Note that Lemma 2 constrains the residual error $\mathbf{e}$ in the over-parameterized regime: $d_0 d_1 \geq N$. In this case eq. (3.2) implies $\mathbf{e} = 0$, if $\mathrm{rank}\,(\mathbf{A} \circ \mathbf{X}) = N$. Therefore, we must have $\mathrm{rank}\,(\mathbf{A} \circ \mathbf{X}) < N$ for sub-optimal DLMs to exist. Later, we use similar rank-based constraints to bound the volume of differentiable regions which contain DLMs with high error. Next, we define this volume formally.

**Angular Volume.** From its definition (eq. (3.1)) each region $\mathcal{D}_{\mathbf{A}}(\mathbf{X})$ has an infinite volume in $\mathbb{R}^{d_1 \times d_0}$: if we multiply a row of $\mathbf{W}$ by a positive scalar, we remain in the same region. Only by rotating the rows of $\mathbf{W}$ can we move between regions. We measure this "angular volume" of a region in a probabilistic way: we randomly sample the rows of $\mathbf{W}$ from an isotropic distribution, *e.g.*, standard Gaussian: $\mathbf{W} \sim \mathcal{N}$, and measure the probability to fall in $\mathcal{D}_{\mathbf{A}}(\mathbf{X})$, arriving to the following

**Definition 3.** For any region $\mathcal{R} \subset \mathbb{R}^{d_1 \times d_0}$. The *angular volume* of $\mathcal{R}$ is

$$\mathcal{V}(\mathcal{R}) \triangleq \mathbb{P}_{\mathbf{W} \sim \mathcal{N}}(\mathbf{W} \in \mathcal{R})\,. \tag{3.3}$$

## 4 MAIN RESULTS

Some of the DLMs are global minima, in which $\mathbf{e} = 0$ and so, $\mathrm{MCE} = \mathrm{MSE} = 0$, while other DLMs are sub-optimal local minima in which $\mathrm{MCE} > \epsilon > 0$. We would like to compare the angular volume (definition 3) corresponding to both types of DLMs. Thus, we make the following definitions.

**Definition 4.** We define[3] $\mathcal{L}_\epsilon \subset \mathbb{R}^{d_1 \times d_0}$ as the union of differentiable regions containing sub-optimal DLMs with $\mathrm{MCE} > \epsilon$ , and $\mathcal{G} \subset \mathbb{R}^{d_1 \times d_0}$ as the union of differentiable regions containing global minima with $\mathrm{MCE} = 0$.

**Definition 5.** We define the constant $\gamma_\epsilon$ as $\gamma_\epsilon \triangleq 0.23 \max\left[\lim_{N \to \infty}(d_0(N)/N), \epsilon\right]^{3/4}$ if $\rho \neq \{0, 1\}$, and $\gamma_\epsilon \triangleq 0.23\epsilon^{3/4}$ if $\rho = 0$.

In this section, we use assumptions 1-4 (stated in section 1) to bound the angular volume of the region $\mathcal{L}_\epsilon$ encapsulating all sub-optimal DLMs, the region $\mathcal{G}$, encapsulating all global minima, and the ratio between the two.

**Angular volume of sub-optimal DLMs.** First, in appendix section 8 we prove the following upper bound in expectation

**Theorem 6.** *Given assumptions 1-4, the expected angular volume of sub-optimal DLMs, with $\mathrm{MCE} > \epsilon > 0$, is exponentially vanishing in $N$ as*

$$\mathbb{E}_{\mathbf{X} \sim \mathcal{N}} \mathcal{V}\left(\mathcal{L}_\epsilon(\mathbf{X}, \mathbf{y})\right) \dot{\leq} \exp\left(-\gamma_\epsilon N^{3/4}\left[d_1 d_0\right]^{1/4}\right)\,.$$

and, using Markov inequality, its immediate probabilistic corollary

**Corollary 7.** *Given assumptions 1-4, for any $\delta > 0$ (possibly a vanishing function of $N$), we have, with probability $1 - \delta$, that the angular volume of sub-optimal DLMs, with $\mathrm{MCE} > \epsilon > 0$, is exponentially vanishing in $N$ as*

$$\mathcal{V}\left(\mathcal{L}_\epsilon(\mathbf{X}, \mathbf{y})\right) \dot{\leq} \frac{1}{\delta} \exp\left(-\gamma_\epsilon N^{3/4}\left[d_1 d_0\right]^{1/4}\right)$$

---

[3]More formally: if $\mathcal{A}(\mathbf{X}, \mathbf{y}, \epsilon)$ is the set of $\mathbf{A} \in \{\rho, 1\}^{d_1 \times N}$ for which $\mathcal{D}_{\mathbf{A}}(\mathbf{X})$ contains a DLM with $\mathrm{MCE} = \epsilon$, then $\forall \epsilon > 0, \mathcal{L}_\epsilon(\mathbf{X}, \mathbf{y}) \triangleq \bigcup_{\epsilon' \geq \epsilon}\left[\bigcup_{\mathbf{A} \in \mathcal{A}(\mathbf{X}, \mathbf{y}, \epsilon')} \mathcal{D}_{\mathbf{A}}(\mathbf{X})\right]$ and $\mathcal{G}(\mathbf{X}, \mathbf{y}) \triangleq \bigcup_{\mathbf{A} \in \mathcal{A}(\mathbf{X}, \mathbf{y}, 0)} \mathcal{D}_{\mathbf{A}}(\mathbf{X})$.

*Proof idea of Theorem* 6: we first show that in differentiable regions with $\mathrm{MCE} > \epsilon > 0$, the condition in Lemma 2, $(\mathbf{A} \circ \mathbf{X}) \mathbf{e} = 0$, implies that $\mathbf{A} = a(\mathbf{WX})$ must have a low rank. Then, we show that, when $\mathbf{X} \sim \mathcal{N}$ and $\mathbf{W} \sim \mathcal{N}$, the matrix $\mathbf{A} = a(\mathbf{WX})$ has a low rank with exponentially low probability. Combining both facts, we obtain the bound.

**Existence of global minima.** Next, to compare the volume of sub-optimal DLMs with that of global minima, in appendix section 9 we show first that, generically, global minima do exist (using a variant of the proof of (Baum, 1988, Theorem 1)):

**Theorem 8.** *For any* $\mathbf{y} \in \{0,1\}^N$ *and* $\mathbf{X} \in \mathbb{R}^{d_0 \times N}$ *almost everywhere[4] we find matrices* $\mathbf{W}^* \in \mathbb{R}^{d_1^* \times d_0}$ *and* $\mathbf{z}^* \in \mathbb{R}^{d_1^*}$, *such that* $\mathbf{y} = f(\mathbf{W}^* \mathbf{X})^\top \mathbf{z}^*$, *where* $d_1^* \triangleq 4 \lceil N/(2d_0 - 2) \rceil$ *and* $\forall i, n$: $\mathbf{w}_i^\top \mathbf{x}^{(n)} \neq 0$. *Therefore, every MNN with* $d_1 \geq d_1^*$ *has a DLM which achieves zero error* $\mathbf{e} = 0$.

Recently (Zhang et al., 2017a, Theorem 1) similarly proved that a 2-layer MNN with approximately $2N$ parameters can achieve zero error. However, that proof required $N$ neurons (similarly to (Nilsson, 1965; Baum, 1988; Yu, 1992; Huang et al., 2006; Livni et al., 2014; Shen, 2016)), while Theorem 8 here requires much less: approximately $d_1^* \approx 2N/d_0$. Also, (Hardt & Ma, 2017, Theorem 3.2) showed a deep residual network with $N \log N$ parameters can achieve zero error. In contrast, here we require just one hidden layer with $2N$ parameters.

Note the construction in Theorem 8 here achieves zero training error by overfitting to the data realization, so it is not expected to be a "good" solution in terms of generalization. To get good generalization, one needs to add additional assumptions on the data ($\mathbf{X}$ and $\mathbf{y}$). Such a possible (common yet insufficient for MNNs) assumption is that the problem is "realizable", *i.e.*, there exist a small "solution MNN", which achieves low error. For example, in the zero error case:

**Assumption 5. (Optional)** *The labels are generated by some teacher* $\mathbf{y} = f(\mathbf{W}^* \mathbf{X})^\top \mathbf{z}^*$ *with weight matrices* $\mathbf{W}^* \in \mathbb{R}^{d_1^* \times d_0}$ *and* $\mathbf{z}^* \in \mathbb{R}^{d_1^*}$ *independent of* $\mathbf{X}$, *for some* $d_1^* \dot{<} N/d_0$.

This assumption is not required for our main result (Theorem 10) – it is merely helpful in improving the following lower bound on $\mathcal{V}(\mathcal{G})$.

**Angular volume of global minima.** We prove in appendix section 10:

**Theorem 9.** *Given assumptions 1-3, we set* $\delta \dot{=} \sqrt{\frac{8}{\pi}} d_0^{-1/2} + 2d_0^{1/2} \sqrt{\log d_0}/N$ *and* $d_1^* = 2N/d_0$, *or if assumption 5 holds, we set* $d_1^*$ *as in this assumption. Then, with probability* $1 - \delta$, *the angular volume of global minima is lower bounded as,*

$$\mathcal{V}(\mathcal{G}(\mathbf{X}, \mathbf{y})) \dot{>} \exp(-d_1^* d_0 \log N) \dot{\geq} \exp(-2N \log N).$$

*Proof idea:* First, we lower bound $\mathcal{V}(\mathcal{G})$ with the angular volume of a single differentiable region of one global minimum $(\mathbf{W}^*, \mathbf{z}^*)$ – either from Theorem 8, or from assumption 5. Then we show that this angular volume is lower bounded when $\mathbf{W} \sim \mathcal{N}$, given a certain angular margin between the datapoints in $\mathbf{X}$ and the rows of $\mathbf{W}^*$. We then calculate the probability of obtaining this margin when $\mathbf{X} \sim \mathcal{N}$. Combining both results, we obtain the final bound.

**Main result: angular volume ratio.** Finally, combining Theorems 6 and 9 it is straightforward to prove our main result in this paper, as we do in appendix section 11:

**Theorem 10.** *Given assumptions 1-3, we set* $\delta \dot{=} \sqrt{\frac{8}{\pi}} d_0^{-1/2} + 2d_0^{1/2} \sqrt{\log d_0}/N$. *Then, with probability* $1 - \delta$, *the angular volume of sub-optimal DLMs, with* $\mathrm{MCE} > \epsilon > 0$, *is exponentially vanishing in N, in comparison to the angular volume of global minima with* $\mathrm{MCE} = 0$

$$\frac{\mathcal{V}(\mathcal{L}_\epsilon(\mathbf{X}, \mathbf{y}))}{\mathcal{V}(\mathcal{G}(\mathbf{X}, \mathbf{y}))} \dot{\leq} \exp\left(-\gamma_\epsilon N^{3/4} [d_1 d_0]^{1/4}\right) \dot{\leq} \exp(-\gamma_\epsilon N \log N).$$

## 5 NUMERICAL EXPERIMENTS

Theorem 10 implies that, with "asymptotically mild" over-parameterization (*i.e.* in which #parameters $= \tilde{\Omega}(N)$), differentiable regions in weight space containing sub-optimal DLMs (with high MCE) are

---

[4]*i.e.*, the set of entries of $\mathbf{X}$, for which the following statement does not hold, has zero measure (Lebesgue).

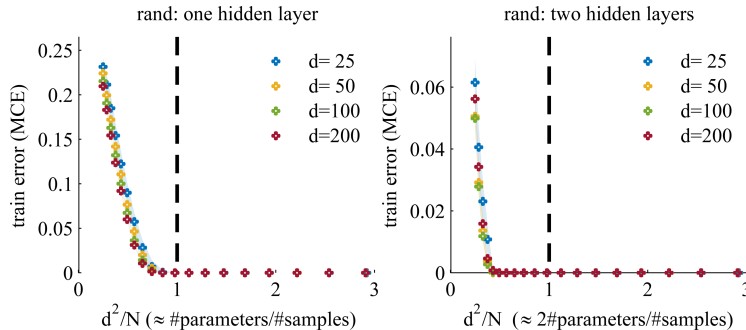

Figure 5.1: **Gaussian data: final training error (mean±std, 30 repetitions) in the over-parameterized regime is low (right of the dashed black line).** We trained MNNs with one and two hiddens layer (with widths equal to $d = d_0$) on a synthetic random dataset in which $\forall n = 1, \ldots, N$, $\mathbf{x}^{(n)}$ was drawn from a normal distribution $\mathcal{N}(0, 1)$, and $y^{(n)} = \pm 1$ with probability $0.5$.

| | MCE | $d_0$ | $d_1$ | $N$ | #parameters/$N$ |
|---|---|---|---|---|---|
| MNIST | **0%** | 784 | 89 | $7 \cdot 10^4$ | **0.999** |
| CIFAR | **0%** | 3072 | 16 | $5 \cdot 10^4$ | **0.983** |
| ImageNet (downsampled to $64 \times 64$) | **0.1%** | 12288 | 105 | $128 \cdot 10^4$ | **1.008** |

Table 1: **Binary classification of MNIST, CIFAR and ImageNet: 1-hidden layer achieves very low training error (MCE) with a few hidden neurons, so that** #parameters $\approx d_0 d_1 \approx N$. In ImageNet we downsampled the images to allow input whitening.

exponentially small in comparison with the same regions for global minima. Since these results are asymptotic in $N \to \infty$, in this section we examine it numerically for a finite number of samples and parameters. We perform experiments on random data, MNIST, CIFAR10 and ImageNet-ILSVRC2012. In each experiment, we used ReLU activations ($\rho = 0$), a binary classification target (we divided the original classes to two groups), MSE loss for optimization (eq. (2.3)), and MCE to determine classification error. Additional implementation details are given in appendix part III.

First, on the small synthetic Gaussian random data (matching our assumptions) we perform a scan on various networks and dataset sizes. With either one or two hidden layers (Figure 5.1) , the error goes to zero when the number of non-redundant parameters (approximately $d_0 d_1$) is greater than the number of samples, as suggested by our asymptotic results. Second, on the non-syntehtic datasets, MNIST, CIFAR and ImageNet (In ImageNet we downsampled the images to size $64 \times 64$, to allow input whitening) we only perform a simulation with a single 1-hidden layer MNN for which #parameters $\approx N$ , and again find (Table 1) that the final error is zero (for MNIST and CIFAR) or very low (ImageNet).

Lastly, in Figure 5.2 we find that, on the Gaussian dataset, the inputs to the hidden neurons converge to a distinctly non-zero value. This indicates we converged to *differentiable* critical points – since non-differentiable critical points must have zero neural inputs. Note that occasionally, during optimization, we could find some neural inputs with very low values near numerical precision level, so convergence to non-differentiable minima may be possible. However, as explained in the next section, as long as the number of neural inputs equal to zero are not too large, our bounds also hold for these minima.

## 6 DISCUSSION

In this paper we examine Differentiable Local Minima (DLMs) of the empiric loss of Multilayer Neural Networks (MNNs) with one hidden layer, scalar output, and LReLU nonlinearities (section 2). We prove (Theorem 10) that with high probability the angular volume (definition 3) of sub-optimal DLMs is exponentially vanishing in comparison to the angular volume of global minima (definition 4), under assumptions 1-4. This results from an upper bound on sub-optimal DLMs (Theorem 6) and a lower bound on global minima (Theorem 9).

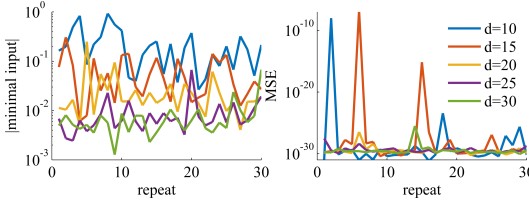

Figure 5.2: **Gaussian data: convergence of the MSE to *differentiable* critical points, as indicated by the convergence of the neural inputs to distinctly non-zero values.** We trained MNNs with one hidden layer on the Gaussian dataset from Figure 5.1, with various widths $d = d_0 = d_1$ and $N = \lfloor d^2/5 \rfloor$ for 1000 epochs, then decreased the learning rate exponentially for another 1000 epochs. This was repeated 30 times. For all $d$ and repeats, we see that *(left)* the final absolute value of the minimal neural input (*i.e.*, $\min_{i,n} \left| \mathbf{w}_i^\top \mathbf{x}^{(n)} \right|$) in the range of $10^{-3} - 10^0$, which is much larger then (r*ight*) the final MSE error for all $d$ and all repeats – in the range $10^{-31} - 10^{-7}$.

**Convergence of SGD to DLMs.** These results suggest a mechanism through which low training error is obtained in such MNNs. However, they do not guarantee it. One issue is that sub-optimal DLMs may have exponentially large basins of attraction. We see two possible paths that might address this issue in future work, using additional assumptions on $\mathbf{y}$. One approach is to show that, with high probability, *no sub optimal DLM* falls within the vanishingly small differentiable regions we bounded in Theorem 6. Another approach would be to bound the size of these basins of attraction, by showing that sufficiently large of number of differentiable regions near the DLM are also vanishingly small (other methods might also help here (Freeman & Bruna, 2016)). Another issue is that SGD might get stuck near differentiable saddle points, if their Hessian does not have strictly negative eigenvalues (*i.e.*, the strict saddle property (Sun et al., 2015)). It should be straightforward to show that such points also have exponentially vanishing angular volume, similar to sub-optimal DLMs. Lastly, SGD might also converge to non-differentiable critical points, which we discuss next.

**Non-differentiable critical points.** The proof of Theorem 6 stems from a first order necessary condition (Lemma 2): $(\mathbf{A} \circ \mathbf{X}) \mathbf{e} = 0$, which is true for any DLM. However, non-differentiable critical points, in which some neural inputs are exactly zero, may also exist (though, numerically, they don't seem very common – see Figure 5.2). In this case, to derive a similar bound, we can replace the condition with $\mathbf{P} (\mathbf{A} \circ \mathbf{X}) \mathbf{e} = 0$, where $\mathbf{P}$ is a projection matrix to the subspace orthogonal to the non-differentiable directions. As long as there are not too many zero neural inputs, we should be able to obtain similar results. For example, if only a constant ratio $r$ of the neural inputs are zero, we can simply choose $\mathbf{P}$ to remove all rows of $(\mathbf{A} \circ \mathbf{X})$ corresponding to those neurons, and proceed with exactly the same proof as before, with $d_1$ replaced with $(1 - r) d_1$. It remains a theoretical challenge to find reasonable assumptions under which the number of non-differentiable directions (*i.e.*, zero neural inputs) does not become too large.

**Related results.** Two works have also derived related results using the $(\mathbf{A} \circ \mathbf{X}) \mathbf{e} = 0$ condition from Lemma 2. In (Soudry & Carmon, 2016), it was noticed that an infinitesimal perturbation of $\mathbf{A}$ makes the matrix $\mathbf{A} \circ \mathbf{X}$ full rank with probability 1 (Allman et al., 2009, Lemma 13) – which entails that $\mathbf{e} = 0$ at all DLMs. Though a simple and intuitive approach, such an infinitesimal perturbation is problematic: from continuity, it cannot change the original MSE at sub-optimal DLMs – unless the weights go to infinity, or the DLM becomes non-differentiable – which are both undesirable results. An extension of this analysis was also done to constrain $\mathbf{e}$ using the singular values of $\mathbf{A} \circ \mathbf{X}$ (Xie et al., 2016), deriving bounds that are easier to combine with generalization bounds. Though a promising approach, the size of the sub-optimal regions (where the error is high) does not vanish exponentially in the derived bounds. More importantly, these bounds require assumptions on the activation kernel spectrum $\gamma_m$, which do not appear to hold in practice (*e.g.*, (Xie et al., 2016, Theorems 1,3) require $m\gamma_m \gg 1$ to hold with high probability, while $m\gamma_m < 10^{-2}$ in (Xie et al., 2016, Figure 1)).

**Modifications and extensions.** There are many relatively simple extensions of these results: the Gaussian assumption could be relaxed to other near-isotropic distributions (*e.g.*, sparse-land model, (Elad, 2010, Section 9.2)) and other convex loss functions are possible instead of the quadratic loss. More challenging directions are extending our results to MNNs with multi-output and multiple hidden layers, or combining our training error results with novel generalization bounds which might be better suited for MNNs (*e.g.*, (Feng et al., 2016; Sokolic et al., 2016; Dziugaite & Roy, 2017)) than previous approaches (Zhang et al., 2017a).

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

# Supplementary information - Appendix

The appendix is divided into three parts. In part I we prove all the main theorems mentioned in the paper. Some of these rely on other technical results, which we prove later in part II. Lastly, in part III we give additional numerical details and results. First, however, we define additional notation (some already defined in the main paper) and mention some known results, which we will use in our proofs.

EXTENDED PRELIMINARIES

- The indicator function $\mathcal{I}(\mathcal{A}) \triangleq \begin{cases} 1 & \text{, if } \mathcal{A} \\ 0 & \text{, else} \end{cases}$, for any event $\mathcal{A}$.

- Kronecker's delta $\delta_{ij} \triangleq \mathcal{I}(i = j)$.

- The Matrix $\mathbf{I}_d$ as the identity matrix in $\mathbb{R}^{d \times d}$, and $\mathbf{I}_{d \times k}$ is the relevant $\mathbb{R}^{d \times k}$ upper left sub-matrix of the identity matrix.

- $[L] \triangleq \{1, 2, \ldots, L\}$

- The vector $\mathbf{m}_n$ as the $n$'th column of a matrix $\mathbf{M}$, unless defined otherwise (then $\mathbf{m}_n$ will be a row of $\mathbf{M}$).

- $\mathbf{M} > 0$ implies that $\forall i, j : M_{ij} > 0$.

- $\mathbf{M}_S$ is the matrix composed of the columns of $\mathbf{M}$ that are in the index set $S$.

- A property holds "$\mathbf{M}$-almost everywhere" (a.e. for short), if the set of entries of $\mathbf{M}$ for which the property does not hold has zero measure (Lebesgue).

- $\|\mathbf{v}\|_0 = \sum_{i=1}^d \mathcal{I}(v_i > 0)$ is the $L_0$ "norm" that counts the number of non-zero values in $\mathbf{v} \in \mathbb{R}^d$.

- If $\mathbf{x} \sim \mathcal{N}(\boldsymbol{\mu}, \boldsymbol{\Sigma})$ the $\mathbf{x}$ is random Gaussian vector.

- $\phi(x) \triangleq \frac{1}{\sqrt{2\pi}} \exp\left(-\frac{1}{2} x^2\right)$ as the univariate Gaussian probability density function.

- $\Phi(x) \triangleq \int_{-\infty}^{x} \phi(u)\, du$ as the Gaussian cumulative distribution function.

- $B(x, y)$ as the beta function.

Lastly, we recall the well known Markov Inequality:

**Fact 11. (Markov Inequality)** *For any random variable $X \geq 0$, we have $\forall \eta > 0$*

$$\mathbb{P}(X \geq \eta) \leq \frac{\mathbb{E}X}{\eta}\,.$$

# Part I

# Proofs of the main results

## 7   FIRST ORDER CONDITION: PROOF OF LEMMA *2*

**Lemma 12. (Lemma 2 restated)** *At all DLMs in $\mathcal{D}_\mathbf{A}(\mathbf{X})$ the residual error $\mathbf{e}$ is identical, and furthermore*

$$(\mathbf{A} \circ \mathbf{X})\mathbf{e} = 0\,. \tag{7.1}$$

*Proof.* Let $\mathbf{W} = [\mathbf{w}_1, \ldots, \mathbf{w}_{d_1}]^\top \in \mathcal{D}_\mathbf{A}(\mathbf{X})$, $\mathbf{G} \triangleq \mathbf{A} \circ \mathbf{X} \in \mathbb{R}^{d_0 d_1 \times N}$, $\tilde{\mathbf{W}} = \mathrm{diag}(\mathbf{z})\mathbf{W} = [\tilde{\mathbf{w}}_1, \ldots, \tilde{\mathbf{w}}_{d_1}]^\top$ and $\tilde{\mathbf{w}} \triangleq \mathrm{vec}\left(\tilde{\mathbf{W}}^\top\right) \in \mathbb{R}^{d_0 d_1}$, where $\mathrm{diag}(\mathbf{v})$ is the diagonal matrix with $\mathbf{v}$ in its

diagonal, and $\text{vec}(\mathbf{M})$ is vector obtained by stacking the columns of the matrix $\mathbf{M}$ on top of one another. Then, we can re-write the MSE (eq. (2.3)) as

$$\text{MSE} = \frac{1}{N} \left\| \mathbf{y} - \mathbf{G}^\top \tilde{\mathbf{w}} \right\|^2 = \frac{1}{N} \left\| \mathbf{e} \right\|^2, \tag{7.2}$$

where $\mathbf{G}^\top \tilde{\mathbf{w}}$ is the output of the MNN. Now, if $(\mathbf{W}, \mathbf{z})$ is a DLM of the MSE in eq. (2.3), then there is no infinitesimal perturbation of $(\mathbf{W}, \mathbf{z})$ which reduces this MSE.

Next, for each row $i$, we will show that $\partial \text{MSE}/\partial \tilde{\mathbf{w}}_i = 0$, since otherwise we can find an infinitesimal perturbation of $(\mathbf{W}, \mathbf{z})$ which decreases the MSE, contradicting the assumption that $(\mathbf{W}, \mathbf{z})$ is a local minimum. For each row $i$, we divide into two cases:

First, we consider the case $z_i \neq 0$. In this case, any infinitesimal perturbation $\mathbf{q}_i$ in $\tilde{\mathbf{w}}_i$ can be produced by an infinitesimal perturbation in $\mathbf{w}_i$: $\tilde{\mathbf{w}}_i + \mathbf{q}_i = (\mathbf{w}_i + \mathbf{q}_i/z_i)z_i$. Therefore, unless the gradient $\partial \text{MSE}/\partial \tilde{\mathbf{w}}_i$ is equal to zero, we can choose an infinitesimal perturbation $\mathbf{q}_i$ in the opposite direction to this gradient, which will decrease the MSE.

Second, we consider the case $z_i = 0$. In this case, the MSE is not affected by changes made exclusively to $\mathbf{w}_i$. Therefore, all $\mathbf{w}_i$ derivatives of the MSE are equal to zero ($\partial^k \text{MSE}/\partial^k \mathbf{w}_i$, to any order $k$). Also, since we are at a differentiable local minimum, $\partial \text{MSE}/\partial z_i = 0$. Thus, using a Taylor expansion, if we perturb $(\mathbf{w}_i, z_i)$ by $(\hat{\mathbf{w}}_i, \hat{z}_i)$ then the MSE is perturbed by

$$\hat{z}_i \hat{\mathbf{w}}_i^\top \frac{\partial}{\partial \tilde{\mathbf{w}}_i} \frac{\partial}{\partial z_i} \text{MSE} + O(\hat{z}_i^2)$$

Therefore, unless $\partial^2 \text{MSE}/(\partial \mathbf{w}_i \partial z_i) = 0$ we can choose $\hat{\mathbf{w}}_i$ and a sufficiently small $\hat{z}_i$ such that the MSE is decreased. Lastly, using the chain rule

$$\frac{\partial}{\partial z_i} \frac{\partial}{\partial \mathbf{w}_i} \text{MSE} = \frac{\partial}{\partial z_i} \left[ z_i \frac{\partial}{\partial \tilde{\mathbf{w}}_i} \text{MSE} \right] = \frac{\partial}{\partial \tilde{\mathbf{w}}_i} \text{MSE}.$$

Thus, $\partial \text{MSE}/\partial \tilde{\mathbf{w}}_i = 0$. This implies that $\tilde{\mathbf{w}}$ is also a DLM[5] of eq. (7.2), which entails

$$0 = -\frac{N}{2} \frac{\partial}{\partial \tilde{\mathbf{w}}_i} \text{MSE} = \mathbf{G} \left( \mathbf{y} - \mathbf{G}^\top \tilde{\mathbf{w}} \right). \tag{7.3}$$

Since $\mathbf{G} = \mathbf{A} \circ \mathbf{X}$ and $\mathbf{e} = \mathbf{y} - \mathbf{G}^\top \tilde{\mathbf{w}}$ this proves eq. (7.1). Now, for any two solutions $\tilde{\mathbf{w}}_1$ and $\tilde{\mathbf{w}}_2$ of eq. (7.3), we have

$$0 = \mathbf{G} \left( \mathbf{y} - \mathbf{G}^\top \tilde{\mathbf{w}}_1 \right) - \mathbf{G} \left( \mathbf{y} - \mathbf{G}^\top \tilde{\mathbf{w}}_1 \right) = \mathbf{G} \mathbf{G}^\top \left( \tilde{\mathbf{w}}_2 - \tilde{\mathbf{w}}_1 \right).$$

Multiplying by $\left( \tilde{\mathbf{w}}_2 - \tilde{\mathbf{w}}_1 \right)^\top$ from the left we obtain

$$\left\| \mathbf{G}^\top \left( \tilde{\mathbf{w}}_2 - \tilde{\mathbf{w}}_1 \right) \right\|^2 = 0 \Rightarrow \mathbf{G}^\top \left( \tilde{\mathbf{w}}_2 - \tilde{\mathbf{w}}_1 \right) = 0.$$

Therefore, the MNN output and the residual error $\mathbf{e}$ are equal for all DLMs in $\mathcal{D}_{\mathbf{A}}(\mathbf{X})$. $\qquad\square$

## 8 SUB-OPTIMAL DIFFERENTIABLE LOCAL MINIMA: PROOF OF THEOREM 6 AND ITS COROLLARY

**Theorem 13.** (**Theorem 6 restated**) *Given assumptions 1-4, the expected angular volume of sub-optimal DLMs, with* $\text{MCE} > \epsilon > 0$*, is exponentially vanishing in $N$ as*

$$\mathbb{E}_{\mathbf{X} \sim \mathcal{N}} \mathcal{V} \left( \mathcal{L}_\epsilon \left( \mathbf{X}, \mathbf{y} \right) \right) \lesssim \exp \left( -\gamma_\epsilon N^{3/4} \left[ d_1 d_0 \right]^{1/4} \right),$$

*where* $\gamma_\epsilon \triangleq 0.23 \max \left[ \lim_{N \to \infty} \left( d_0 \left( N \right) / N \right), \epsilon \right]^{3/4}$ *if $\rho \neq \{0, 1\}$, and* $\gamma_\epsilon \triangleq 0.23 \epsilon^{3/4}$ *if $\rho = 0$.*

To prove this theorem we upper bound the angular volume of $\mathcal{L}_\epsilon$ (definition 4), *i.e.*, differentiable regions in which there exist DLMs with $\text{MCE} > \epsilon > 0$. Our proof uses the first order necessary condition for DLMs from Lemma 2, $(\mathbf{A} \circ \mathbf{X}) \mathbf{e} = 0$, to find which configurations of $\mathbf{A}$ allow for

---

[5]Note that the converse argument is not true – a DLM in $\tilde{\mathbf{w}}$ might not be a DLM in $(\mathbf{W}, \mathbf{z})$.

a high residual error $\mathbf{e}$ with $\mathrm{MCE} > \epsilon > 0$. In these configurations $\mathbf{A} \circ \mathbf{X}$ cannot have full rank, and therefore, as we show (Lemma 14 below), $\mathbf{A} = a(\mathbf{WX})$ must have a low rank. However, $\mathbf{A} = a(\mathbf{WX})$ has a low rank with exponentially low probability when $\mathbf{X} \sim \mathcal{N}$ and $\mathbf{W} \sim \mathcal{N}$ (Lemmas 15 and 16 below). Thus, we derive an upper bound on $\mathbb{E}_{\mathbf{X} \sim \mathcal{N}} \mathcal{V}(\mathcal{L}_\epsilon(\mathbf{X}, \mathbf{y}))$.

Before we begin, let us recall some notation: $[L] \triangleq \{1, 2, \ldots, L\}$, $\mathbf{M} > 0$ implies that $\forall i, j : M_{ij} > 0$, $\mathbf{M}_S$ is the matrix composed of the columns of $\mathbf{M}$ that are in the index set $S$, $\|\mathbf{v}\|_0$ as the $L_0$ "norm" that counts the number of non-zero values in $\mathbf{v}$. First we consider the case $\rho \neq 0$. Also, we denote $K_r \triangleq \max[N\epsilon, rd_0]$.

First we consider the case $\rho \neq 0$.

From definition 3 of the angular volume

$$
\begin{aligned}
\mathbb{E}_{\mathbf{X} \sim \mathcal{N}} \mathcal{V}(\mathcal{L}_\epsilon(\mathbf{X}, \mathbf{y})) &= \mathbb{P}_{(\mathbf{X}, \mathbf{y}) \sim \mathbb{P}_{X,Y}, \mathbf{W} \sim \mathcal{N}}(\mathbf{W} \in \mathcal{L}_\epsilon(\mathbf{X}, \mathbf{y})) \\
&\overset{(1)}{\leq} \mathbb{P}_{(\mathbf{X}, \mathbf{y}) \sim \mathbb{P}_{X,Y}, \mathbf{W} \sim \mathcal{N}}\left(\exists \mathbf{A} \in \{\rho, 1\}^{d_1 \times N}, \, \mathbf{W} \in \mathcal{D}_\mathbf{A}(\mathbf{X}), \, \mathbf{v} \in \mathbb{R}^N : (\mathbf{A} \circ \mathbf{X})\mathbf{v} = 0, N\epsilon \leq \|\mathbf{v}\|_0\right) \\
&\overset{(2)}{=} \mathbb{P}_{\mathbf{X} \sim \mathcal{N}, \mathbf{W} \sim \mathcal{N}}\left(\exists \mathbf{A} \in \{\rho, 1\}^{d_1 \times N}, \, \mathbf{W} \in \mathcal{D}_\mathbf{A}(\mathbf{X}), \, \mathbf{v} \in \mathbb{R}^N : (\mathbf{A} \circ \mathbf{X})\mathbf{v} = 0, N\epsilon \leq \|\mathbf{v}\|_0\right) \\
&\overset{(3)}{\leq} \mathbb{P}_{\mathbf{X} \sim \mathcal{N}, \mathbf{W} \sim \mathcal{N}}(\exists S \subset [N] : |S| \geq \max[N\epsilon, \mathrm{rank}(a(\mathbf{WX}_S))d_0 + 1]) \\
&= \mathbb{E}_{\mathbf{X} \sim \mathcal{N}}[\mathbb{P}_{\mathbf{W} \sim \mathcal{N}}(\exists S \subset [N] : |S| \geq \max[N\epsilon, \mathrm{rank}(a(\mathbf{WX}_S))d_0 + 1]|\mathbf{X})] \\
&\overset{(4)}{\leq} \mathbb{E}_{\mathbf{X} \sim \mathcal{N}}\left[\sum_{r=1}^{N/d_0} \mathbb{P}_{\mathbf{W} \sim \mathcal{N}}(\exists S \subset [N] : |S| = K_r, \mathrm{rank}(a(\mathbf{WX}_S)) = r|\mathbf{X})\right] \\
&\overset{(5)}{\leq} \mathbb{E}_{\mathbf{X} \sim \mathcal{N}}\left[\sum_{r=1}^{N/d_0} \sum_{S:|S|=K_r} \mathbb{P}_{\mathbf{W} \sim \mathcal{N}}(\mathrm{rank}(a(\mathbf{WX}_S)) = r|\mathbf{X})\right], \quad\quad (8.1)
\end{aligned}
$$

where

1. If we are at DLM $a$ in $\mathcal{D}_\mathbf{A}(\mathbf{X})$, then Lemma 2 implies $(\mathbf{A} \circ \mathbf{X})\mathbf{e} = 0$. Also, if $e^{(n)} = 0$ on some sample, we necessarily classify it correctly, and therefore $\mathrm{MCE} \leq \|\mathbf{e}\|_0 / N$. Since $\mathrm{MCE} > \epsilon$ in $\mathcal{L}_\epsilon$ this implies that $N\epsilon < \|\mathbf{e}\|_0$. Thus, this inequality holds for $\mathbf{v} = \mathbf{e}$.

2. We apply assumption 1, that $\mathbf{X} \sim \mathcal{N}$.

3. Assumption 4 implies $d_0 d_1 \dot{>} N \log^4 N \geq N$. Thus, we can apply the following Lemma, proven in appendix section 12.1:

   **Lemma 14.** *Let* $\mathbf{X} \in \mathbb{R}^{d_0 \times N}$, $\mathbf{A} \in \{\rho, 1\}^{d_1 \times N}$, $S \subset [N]$ *and* $d_0 d_1 \geq N$. *Then, simultaneously for every possible* $\mathbf{A}$ *and* $S$ *such that*

   $$|S| \leq \mathrm{rank}(\mathbf{A}_S)d_0,$$

   *we have that,* $\mathbf{X}$-*a.e.,* $\nexists \mathbf{v} \in \mathbb{R}^N$ *such that* $v_n \neq 0 \, \forall n \in S$ *and* $(\mathbf{A} \circ \mathbf{X})\mathbf{v} = 0$.

4. Recall that $K_r \triangleq \max[N\epsilon, rd_0]$. We use the union bound over all possible ranks $r \geq 1$: we ignore the $r = 0$ case since for $\rho \neq 0$ (see eq. (2.1)) there is zero probability that $\mathrm{rank}(a(\mathbf{WX}_S)) = 0$ for some non-empty $S$. For each rank $r \geq 1$, it is required that $|S| > K_r = \max[N\epsilon, rd_0]$, so $|S| = K_r$ is a relaxation of the original condition, and thus its probability is not lower.

5. We again use the union bound over all possible subsets $S$ of size $K_r$.

Thus, from eq. (8.1), we have

$$\mathbb{E}_{\mathbf{X} \sim \mathcal{N}} \mathcal{V} \left( \mathcal{L}_\epsilon \left( \mathbf{X}, \mathbf{y} \right) \right)$$

$$\leq \sum_{r=1}^{N/d_0} \sum_{S : |S| = K_r} \mathbb{E}_{\mathbf{X} \sim \mathcal{N}} \left[ \mathbb{P}_{\mathbf{W} \sim \mathcal{N}} \left( \mathrm{rank} \left( a \left( \mathbf{W} \mathbf{X}_S \right) \right) = r | \mathbf{X} \right) \right]$$

$$\overset{(1)}{=} \sum_{r=1}^{N/d_0} \left( \begin{array}{c} N \\ K_r \end{array} \right) \mathbb{P}_{\mathbf{X} \sim \mathcal{N}, \mathbf{W} \sim \mathcal{N}} \left( \mathrm{rank} \left( a \left( \mathbf{W} \mathbf{X}_{[K_r]} \right) \right) = r \right)$$

$$\overset{(2)}{\dot{\leq}} \sum_{r=1}^{N/d_0} \left( \begin{array}{c} N \\ K_r \end{array} \right) 2^{K_r + r d_0 (\log d_1 + \log K_r) + r^2} \mathbb{P}_{\mathbf{X} \sim \mathcal{N}, \mathbf{W} \sim \mathcal{N}} \left( \mathbf{W} \mathbf{X}_{[K_r/2]} > 0 \right)$$

$$\overset{(3)}{\dot{\leq}} \sum_{r=1}^{N/d_0} \left( \begin{array}{c} N \\ K_r \end{array} \right) 2^{K_r + r d_0 (\log d_1 + \log K_r) + r^2} \exp \left( -0.2 K_r \left( 2 \frac{d_0 d_1}{K_r} \right)^{1/4} \right)$$

$$\overset{(4)}{\dot{\leq}} \sum_{r=1}^{N/d_0} 2^{N \log N} \exp \left( -0.23 N^{3/4} \left[ d_1 d_0 \right]^{1/4} \max \left[ \epsilon, r d_0 / N \right]^{3/4} \right)$$

$$\overset{(5)}{\dot{\leq}} \exp \left( -\gamma_\epsilon N^{3/4} \left[ d_1 d_0 \right]^{1/4} \right) . \tag{8.2}$$

1. Since we take the expectation over $\mathbf{X}$, the location of $S$ does not affect the probability. Therefore, we can set without loss of generality $S = [K_r]$.

2. Note that $r \leq N/d_0 \dot{<} \min \left[ d_0, d_1 \right]$ from assumptions 3 and 4. Thus, with $k = K_r \geq d_0$, we apply the following Lemma, proven in appendix section 12.2:

   **Lemma 15.** *Let $\mathbf{X} \in \mathbb{R}^{d_0 \times k}$ be a random matrix with independent and identically distributed columns, and $\mathbf{W} \in \mathbb{R}^{d_1 \times d_0}$ an independent standard random Gaussian matrix. Then, in the limit $\min \left[ k, d_0, d_1 \right] \dot{>} r$,*

   $$\mathbb{P} \left( \mathrm{rank} \left( a \left( \mathbf{W} \mathbf{X} \right) \right) = r \right) \dot{\leq} 2^{k + r d_0 (\log d_1 + \log k) + r^2} \mathbb{P} \left( \mathbf{W} \mathbf{X}_{[\lfloor k/2 \rfloor]} > 0 \right) .$$

3. Note that $K_r \geq N \epsilon \dot{=} N > 2 d_1$, and $\min \left[ K_r, d_0, d_1 \right] \dot{>} d_0 d_1 / K_r \dot{>} 1$ from assumptions 2 and 4. Thus, we apply the following Lemma (with $\mathbf{C} = \mathbf{X}^\top$, $\mathbf{B} = \mathbf{W}^\top$, $M = d_0$, $L = d_1$ and $N = K_r/2$), proven in appendix section 12.3:

   **Lemma 16.** *Let $\mathbf{C} \in \mathbb{R}^{N \times M}$ and $\mathbf{B} \in \mathbb{R}^{M \times L}$ be two independent standard random Gaussian matrices. Without loss of generality, assume $N \geq L$, and denote $\alpha \triangleq ML/N$. Then, in the regime $M \leq N$ and in the limit $\min \left[ N, M, L \right] \dot{>} \alpha \dot{>} 1$, we have*

   $$\mathbb{P} \left( \mathbf{C} \mathbf{B} > 0 \right) \dot{\leq} \exp \left( -0.4 N \alpha^{1/4} \right) .$$

4. We use $r d_0 \leq N$, $\left( \begin{array}{c} N \\ K_r \end{array} \right) \leq 2^N$, $K_r \leq N$, and $d_1 \dot{<} N$ (from assumption 4) and $r^2 \leq N^2 / d_0^2 \dot{<} N$ (from assumption (3)) to simplify the combintaorial expressions.

5. First, note that $r = 1$ is the maximal term in the sum, so we can neglect the other, exponentially smaller, terms. Second, from assumption 3 we have $d_0 \dot{\leq} N$, so

   $$\lim_{N \to \infty} 0.23 \max \left[ \epsilon, d_0 \left( N \right) / N \right]^{3/4} = 0.23 \max \left[ \epsilon, \lim_{N \to \infty} d_0 \left( N \right) / N \right]^{3/4} = \gamma_\epsilon .$$

   Third, from assumption 4 we have $N \log^4 N \dot{<} d_0 d_1$, so the $2^{N \log N}$ term is negligible.

Thus,

$$\mathbb{E}_{\mathbf{X} \sim \mathcal{N}} \mathcal{V} \left( \mathcal{L}_\epsilon \left( \mathbf{X}, \mathbf{y} \right) \right) \dot{\leq} \exp \left( -\gamma_\epsilon N^{3/4} \left[ d_1 d_0 \right]^{1/4} \right) . \tag{8.3}$$

which proves the Theorem for the case $\rho \neq 0$.

Next, we consider the case $\rho = 0$. In this case, we need to change transition (4) in eq. (8.1), so the sum starts from $r = 0$, since now we can have $\mathrm{rank} \left( a \left( \mathbf{W} \mathbf{X}_S \right) \right) = 0$. Following exactly the same

logic (except the modification to the sum), we only need to modify transition (5) in eq. (8.2) – since now the maximal term in the sum is at $r = 0$. This entails $\gamma_\epsilon = 0.23\epsilon^{3/4}$.

∎

**Corollary 17. (Corollary 7 restated)** *Given assumptions 1-4, for any $\delta > 0$ (possibly a vanishing function of $N$), we have, with probability $1 - \delta$, that the angular volume of sub-optimal DLMs, with* MCE $> \epsilon > 0$, *is exponentially vanishing in $N$ as*

$$\mathcal{V}\left(\mathcal{L}_\epsilon\left(\mathbf{X}, \mathbf{y}\right)\right) \dot{\leq} \frac{1}{\delta} \exp\left(-\gamma_\epsilon N^{3/4} \left[d_1 d_0\right]^{1/4}\right)$$

*Proof.* Since $\mathcal{V}\left(\mathcal{L}_\epsilon\left(\mathbf{X}, \mathbf{y}\right)\right) \geq 0$ we can use Markov's Theorem (Fact 11) $\forall \eta > 0$:

$$\mathbb{P}_{\mathbf{X} \sim \mathcal{N}}\left(\mathcal{V}\left(\mathcal{L}_\epsilon\left(\mathbf{X}, \mathbf{y}\right)\right) < \eta\right) > 1 - \frac{\mathbb{E}_{\mathbf{X} \sim \mathcal{N}} \mathcal{V}\left(\mathcal{L}_\epsilon\left(\mathbf{X}, \mathbf{y}\right)\right)}{\eta}$$

denoting $\eta = \frac{1}{\delta}\mathbb{E}_{\mathbf{X} \sim \mathcal{N}} \mathcal{V}\left(\mathcal{L}_\epsilon\left(\mathbf{X}, \mathbf{y}\right)\right)$, and using Theorem (6) we prove the corollary.

$$1 - \delta < \mathbb{P}_{\mathbf{X} \sim \mathcal{N}}\left(\mathcal{V}\left(\mathcal{L}_\epsilon\left(\mathbf{X}, \mathbf{y}\right)\right) < \frac{1}{\delta}\mathbb{E}_{\mathbf{X} \sim \mathcal{N}} \mathcal{V}\left(\mathcal{L}_\epsilon\left(\mathbf{X}, \mathbf{y}\right)\right)\right)$$

$$< \mathbb{P}_{\mathbf{X} \sim \mathcal{N}}\left(\mathcal{V}\left(\mathcal{L}_\epsilon\left(\mathbf{X}, \mathbf{y}\right)\right) \dot{\leq} \frac{1}{\delta} \exp\left(-\gamma_\epsilon N^{3/4} \left[d_1 d_0\right]^{1/4}\right)\right)$$

where we note that replacing a regular inequality $<$ with inequality in the leading order $\dot{\leq}$ only removes constraints, and therefore increases the probability. □

## 9 CONSTRUCTION OF GLOBAL MINIMA: PROOF OF THEOREM 8:

Recall the LReLU non-linearity

$$f\left(x\right) \triangleq \begin{cases} \rho x & , \text{ if } x < 0 \\ x & , \text{ if } x \geq 0 \end{cases}$$

in eq. (2.1), where $\rho \neq 1$.

**Theorem 18. (Theorem 8 restated)** *For any $\mathbf{y} \in \{0,1\}^N$ and $\mathbf{X} \in \mathbb{R}^{d_0 \times N}$ almost everywhere we find matrices $\mathbf{W}^* \in \mathbb{R}^{d_1^* \times d_0}$ and $\mathbf{z}^* \in \mathbb{R}^{d_1^*}$, such that $\mathbf{y} = f\left(\mathbf{W}^*\mathbf{X}\right)^\top \mathbf{z}^*$, where $d_1^* \triangleq 4\lceil N/\left(2d_0 - 2\right)\rceil$ and $\forall i, n: \mathbf{w}_i^\top\mathbf{x}^{(n)} \neq 0$. Therefore, every MNN with $d_1 \geq d_1^*$ has a DLM which achieves zero error $\mathbf{e} = 0$.*

We prove the existence of a solution $(\mathbf{W}^*, \mathbf{z}^*)$, by explicitly constructing it. This construction is a variant of (Baum, 1988, Theorem 1), except we use LReLU without bias and MSE – instead of threshold units with bias and MCE. First, we note that for any $\epsilon_1 > \epsilon_2 > 0$, the following trapezoid function can be written as a scaled sum of four LReLU:

$$\tau\left(x\right) \triangleq \begin{cases} 0 & , \text{ if } |x| > \epsilon_1 \\ 1 & , \text{ if } |x| \leq \epsilon_2 \\ \frac{\epsilon_1 - |x|}{\epsilon_1 - \epsilon_2} & , \text{ if } \epsilon_2 < |x| \leq \epsilon_1 \end{cases} \tag{9.1}$$

$$= \frac{1}{\epsilon_1 - \epsilon_2}\frac{1}{1 - \rho}\left[f\left(x + \epsilon_1\right) - f\left(x + \epsilon_2\right) - f\left(x - \epsilon_2\right) + f\left(x - \epsilon_1\right)\right].$$

Next, we examine the set of data points which are classified to 1: $\mathcal{S}^+ \triangleq \left\{n \in [N] \,|\, y^{(n)} = 1\right\}$. Without loss of generality, assume $|\mathcal{S}^+| \leq \frac{N}{2}$. We partition $\mathcal{S}^+$ to

$$K = \left\lceil \frac{|\mathcal{S}^+|}{d_0 - 1} \right\rceil \leq \left\lceil \frac{N}{2\left(d_0 - 1\right)} \right\rceil$$

subsets $\left\{\mathcal{S}_i^+\right\}_{i=1}^K$, each with no more than $d_0 - 1$ samples. For almost any dataset we can find $K$ hyperplanes passing through the origin, with normals $\{\tilde{\mathbf{w}}_i\}_{i=1}^K$ such that each hyperplane contains all $d_0 - 1$ points in subset $\mathcal{S}_i^+$, *i.e.*,

$$\tilde{\mathbf{w}}_i^\top \mathbf{X}_{\mathcal{S}_i^+} = 0, \tag{9.2}$$

but no other point, so $\forall n \notin \mathcal{S}_i^+ : \tilde{\mathbf{w}}_i^\top \mathbf{x}^{(n)} \neq 0$,

If $\epsilon_1, \epsilon_2$ in eq. (9.1) are sufficiently small ($\forall n \notin \mathcal{S}_i^+ : \left| \tilde{\mathbf{w}}_i^\top \mathbf{x}^{(n)} \right| > \epsilon_1$) then we have

$$
\tau \left( \tilde{\mathbf{w}}_i^\top \mathbf{x}^{(n)} \right) = \begin{cases} 1 & , \text{ if } n \in \mathcal{S}_i^+ \\ 0 & , \text{ else} \end{cases} .
$$

Then we have

$$
\sum_{i=1}^K \tau \left( \tilde{\mathbf{w}}_i^\top \mathbf{x}^{(n)} \right) = \begin{cases} 1 & , \text{ if } n \in \mathcal{S}^+ \\ 0 & , \text{ else} \end{cases} \tag{9.3}
$$

which gives the correct classification on all the data points. Thus, from eq. (9.1), we can construct a MNN with

$$
d_1^* = 4K
$$

hidden neurons which achieves zero error. This is straightforward to do if we have a bias in each neuron. To construct this MNN even without bias, we first find a vector $\hat{\mathbf{w}}_i$ such that

$$
\hat{\mathbf{w}}_i^\top \left[ \mathbf{X}_{\mathcal{S}_i^+}, \tilde{\mathbf{w}}_i \right] = [1, \dots, 1, 1, 0] . \tag{9.4}
$$

Note that this is possible since $\left[ \mathbf{X}_{\mathcal{S}_i^+}, \tilde{\mathbf{w}}_i \right]$ has full rank $\mathbf{X}$-a.e. (the matrix $\mathbf{X}_{\mathcal{S}_i^+} \in \mathbb{R}^{d_0 \times d_0 - 1}$ has, $\mathbf{X}$-a.e., one zero left eigenvector, which is $\tilde{\mathbf{w}}_i$, according to eq. (9.2)). Additionally, we can set

$$
\|\tilde{\mathbf{w}}_i\| = \|\hat{\mathbf{w}}_i\| , \tag{9.5}
$$

since changing the scale of $\mathbf{w}_i$ would not affect the validity of eq. (9.2). Then, we denote

$$
\mathbf{w}_i^{(1)} \triangleq \tilde{\mathbf{w}}_i + \epsilon_1 \hat{\mathbf{w}}_i \;;\; \mathbf{w}_i^{(2)} \triangleq \tilde{\mathbf{w}}_i + \epsilon_2 \hat{\mathbf{w}}_i
$$

$$
\mathbf{w}_i^{(3)} \triangleq \tilde{\mathbf{w}}_i - \epsilon_2 \hat{\mathbf{w}}_i \;;\; \mathbf{w}_i^{(4)} \triangleq \tilde{\mathbf{w}}_i - \epsilon_1 \hat{\mathbf{w}}_i .
$$

Note, from eqs. (9.2) and (9.4) that this choice satisfies

$$
\forall n \in \mathcal{S}_i^+ : \; \mathbf{w}_i^{(j)\top} \mathbf{x}^{(n)} = \begin{cases} \epsilon_1 & , \text{ if } j = 1 \\ \epsilon_2 & , \text{ if } j = 2 \\ -\epsilon_2 & , \text{ if } j = 3 \\ -\epsilon_1 & , \text{ if } j = 4 \end{cases} . \tag{9.6}
$$

Also, to ensure that $\forall n \notin \mathcal{S}_i^+$ the sign of $\mathbf{w}_i^{(j)\top} \mathbf{x}^{(n)}$ does not change for different $j$, for some $\beta, \gamma < 1$ we define

$$
\epsilon_1 = \beta \frac{\min_{n \notin \mathcal{S}_i^+} \left| \tilde{\mathbf{w}}_i^\top \mathbf{x}^{(n)} \right|}{\max_{n \notin \mathcal{S}_i^+} \left| \hat{\mathbf{w}}_i^\top \mathbf{x}^{(n)} \right|} , \epsilon_2 = \gamma \epsilon_1 , \tag{9.7}
$$

where with probability 1, $\min_{n \notin \mathcal{S}_i^+} \left| \tilde{\mathbf{w}}_i^\top \mathbf{x}^{(n)} \right| > 0$ and $\max_{n \notin \mathcal{S}_i^+} \left| \hat{\mathbf{w}}_i^\top \mathbf{x}^{(n)} \right| > 0$. Defining

$$
\mathbf{W}_i \triangleq \left[ \mathbf{w}_i^{(1)}, \mathbf{w}_i^{(2)}, \mathbf{w}_i^{(3)}, \mathbf{w}_i^{(4)} \right]^\top \in \mathbb{R}^{4K \times d_0} \tag{9.8}
$$

$$
\mathbf{z}_i \triangleq [1, -1, -1, 1]^\top \in \mathbb{R}^4
$$

and combining all the above facts, we have

$$
f \left( \mathbf{W}_i \mathbf{x}^{(n)} \right)^\top \mathbf{z}_i
$$
$$
= \frac{1}{\epsilon_1 - \epsilon_2} \frac{1}{1 - \rho} \left[ f \left( \mathbf{w}_i^{(1)\top} \mathbf{x}^{(n)} \right) - f \left( \mathbf{w}_i^{(2)\top} \mathbf{x}^{(n)} \right) - f \left( \mathbf{w}_i^{(3)\top} \mathbf{x}^{(n)} \right) + f \left( \mathbf{w}_i^{(3)\top} \mathbf{x}^{(n)} \right) \right]
$$
$$
= \frac{1}{\epsilon_1 - \epsilon_2} \frac{1}{1 - \rho} \left[ f \left( \tilde{\mathbf{w}}_i^\top \mathbf{x}^{(n)} + \epsilon_1 \hat{\mathbf{w}}_i^\top \mathbf{x}^{(n)} \right) - f \left( \tilde{\mathbf{w}}_i^\top \mathbf{x} + \epsilon_2 \hat{\mathbf{w}}_i^\top \mathbf{x}^{(n)} \right) \right.
$$
$$
\left. - f \left( \tilde{\mathbf{w}}_i^\top \mathbf{x}^{(n)} - \epsilon_2 \hat{\mathbf{w}}_i^\top \mathbf{x}^{(n)} \right) + f \left( \tilde{\mathbf{w}}_i^\top \mathbf{x}^{(n)} - \epsilon_1 \hat{\mathbf{w}}_i^\top \mathbf{x}^{(n)} \right) \right]
$$
$$
= \begin{cases} 1 & , \text{ if } n \in \mathcal{S}_i^+ \\ 0 & , \text{ else} \end{cases} .
$$

Thus, for

$$\mathbf{W}^* = \left[\mathbf{W}_1^\top, \ldots, \mathbf{W}_K^\top\right]^\top \in \mathbb{R}^{4 \times d_0}$$

$$\mathbf{z}^* = \frac{1}{\epsilon_1 - \epsilon_2} \frac{1}{1 - \rho} \cdot [\mathbf{z}_1, \ldots, \mathbf{z}_K] \in \mathbb{R}^{4K}$$

we obtain a MNN that implements

$$f\left(\mathbf{W}^*\mathbf{x}^{(n)}\right)^\top \mathbf{z}^* = \begin{cases} 1 & \text{, if } n \in \mathcal{S}^+ \\ 0 & \text{, else} \end{cases}$$

and thus achieves zero error. Clearly, from this construction, if $\mathbf{w}_i$ is a row of $\mathbf{W}^*$, then $\forall n \in \mathcal{S}_i^+, \forall i : \left|\mathbf{w}_i^\top \mathbf{x}^{(n)}\right| \geq \epsilon_2$, and with probability 1 $\forall n \notin \mathcal{S}_i^+, \forall i : \left|\mathbf{w}_i^\top \mathbf{x}^{(n)}\right| > 0$, so this construction does not touch any non-differentiable region of the MSE. $\blacksquare$

## 10 GLOBAL MINIMA: PROOF OF THEOREM 9

**Theorem 19. (Theorem 9 restated).** *Given assumptions 1-3, we set $\delta \doteq \sqrt{\frac{8}{\pi}} d_0^{-1/2} + 2d_0^{1/2}\sqrt{\log d_0}/N$ and $d_1^* = 2N/d_0$, or if assumption 5 holds, we set $d_1^*$ as in this assumption. Then, with probability $1 - \delta$, the angular volume of global minima is lower bounded as,*

$$\mathcal{V}\left(\mathcal{G}\left(\mathbf{X}, \mathbf{y}\right)\right) \dot{>} \exp\left(-d_1^* d_0 \log N\right) \dot{\geq} \exp\left(-2N \log N\right).$$

In this section we lower bound the angular volume of $\mathcal{G}$ (definition 4), *i.e.*, differentiable regions in which there exist DLMs with MCE = 0. We lower bound $\mathcal{V}(\mathcal{G})$ using the angular volume corresponding to the differentiable region containing a single global minimum.

From assumption 4, we have $d_0 d_1 \dot{>} N$, so we can apply Theorem 8 and say that the labels are generated using a $(\mathbf{X}, \mathbf{y})$ -dependent MNN: $\mathbf{y} = f\left(\mathbf{W}^*\mathbf{X}\right)^\top \mathbf{z}^*$ with target weights $\mathbf{W}^* = \left[\mathbf{w}_1^{*\top}, \ldots, \mathbf{w}_{d_1^*}^{*\top}\right]^\top \in \mathbb{R}^{d_1^* \times d_0}$ and $\mathbf{z}^* \in \mathbb{R}^{d_1}$. If, in addition, assumption 5 holds then we can assume $\mathbf{W}^*$ and $\mathbf{z}^*$ are independent from $(\mathbf{X}, \mathbf{y})$. In both cases, the following differentiable region

$$\tilde{\mathcal{G}}\left(\mathbf{X}, \mathbf{W}^*\right) \triangleq \left\{\mathbf{W} \in \mathbb{R}^{d_1 \times d_0} | \forall i \leq d_1^* : \text{ sign}\left(\mathbf{w}_i^\top \mathbf{X}\right) = \text{sign}\left(\mathbf{w}_i^{*\top}\mathbf{X}\right)\right\}, \quad (10.1)$$

also contains a differentiable global minimum (just set $\mathbf{w}_i = \mathbf{w}_i^*$, $z_i = z_i^*$ $\forall i \leq d_1^*$, and $z_i = 0$ $\forall i > d_1^*$), and therefore $\forall \mathbf{X}, \mathbf{y}$ and their corresponding $\mathbf{W}^*$, we have

$$\mathcal{G}\left(\mathbf{X}, \mathbf{y}\right) \supset \tilde{\mathcal{G}}\left(\mathbf{X}, \mathbf{W}^*\right) \quad (10.2)$$

Also, we will make use of the following definition.

**Definition 20.** Let $\mathbf{X}$ have an *angular margin* $\alpha$ from $\mathbf{W}^*$ if all datapoints (columns in $\mathbf{X}$) are at an angle of at least $\alpha$ from all the weight hyperplanes (rows of $\mathbf{W}^*$), *i.e.*, $\mathbf{X}$ is in the set

$$\mathcal{M}^\alpha\left(\mathbf{W}^*\right) \triangleq \left\{\mathbf{X} \in \mathbb{R}^{d_0 \times N} | \forall i, n : \left|\frac{\mathbf{x}^{(n)\top}\mathbf{w}_i^*}{\|\mathbf{x}^{(n)}\| \|\mathbf{w}_i^*\|}\right| > \sin\alpha\right\}. \quad (10.3)$$

Using the definitions in eqs. (10.3) and (10.1), we prove the Theorem using the following three Lemmas.

First, In appendix section 13.2 we prove

**Lemma 21.** *For any $\alpha$, if $\mathbf{W}^*$ is independent from $\mathbf{W}$ then, in the limit $N \to \infty$, $\forall \mathbf{X} \in \mathcal{M}^\alpha\left(\mathbf{W}^*\right)$ with $\log \sin\alpha \dot{>} d_0^{-1} \log d_0$*

$$\mathcal{V}\left(\tilde{\mathcal{G}}\right) = \mathbb{P}_{\mathbf{W} \sim \mathcal{N}}\left(\mathbf{W} \in \tilde{\mathcal{G}}\left(\mathbf{X}, \mathbf{W}^*\right)\right) \dot{\geq} \exp\left(d_0 d_1^* \log \sin\alpha\right).$$

Second, in appendix section 13.3 we prove

**Lemma 22.** *Let* $\mathbf{W}^* \in \mathbb{R}^{d_1^* \times d_0}$ *a fixed matrix independent of* $\mathbf{X}$. *Then, in the limit* $N \to \infty$ *with* $d_1^* \dot{\leq} d_0 \dot{\leq} N$, *the probability of not having an angular margin* $\sin \alpha = 1/(d_1^* d_0 N)$ *(eq. (10.3)) is upper bounded by*

$$\mathbb{P}\left(\mathbf{X} \notin \mathcal{M}^\alpha\left(\mathbf{W}^*\right)\right) \dot{\leq} \sqrt{\frac{2}{\pi}} d_0^{-1/2}$$

Lastly, in appendix section 13.4 we prove

**Lemma 23.** *Let* $\mathbf{X} \in \mathbb{R}^{d_0 \times N}$ *be a standard random Gaussian matrix of datapoints. Then we can find, with probability 1,* $(\mathbf{X}, \mathbf{y})$*-dependent matrices* $\mathbf{W}^*$ *and* $\mathbf{z}^*$ *as in Theorem 8 (where* $d_1^* \triangleq 4\lceil N/(2d_0 - 2)\rceil$*). Moreover, in the limit* $N \to \infty$*, where* $N/d_0 \dot{\leq} d_0 \dot{\leq} N$*, for any* $\mathbf{y}$*, we can bound the probability of not having an angular margin (eq. (10.3)) with* $\sin \alpha = 1/(d_1^* d_0 N)$ *by*

$$\mathbb{P}\left(\mathbf{X} \notin \mathcal{M}^\alpha\left(\mathbf{W}^*\right)\right) \dot{\leq} \sqrt{\frac{8}{\pi}} d_0^{-1/2} + \frac{2d_0^{1/2}\sqrt{\log d_0}}{N}$$

Recall that $\forall \mathbf{X}, \mathbf{y}$ and their corresponding $\mathbf{W}^*$, we have $\mathcal{G}\left(\mathbf{X}, \mathbf{y}\right) \subset \tilde{\mathcal{G}}\left(\mathbf{X}, \mathbf{W}^*\right)$ (eq. (10.2)). Thus, combining Lemmas 21 with $\sin \alpha = 1/(d_1^* d_0 N)$ together with either Lemma 22 or 23, we prove the first (left) inequality of Theorem 9:

$$\mathcal{V}\left(\mathcal{G}\left(\mathbf{X}, \mathbf{y}\right)\right) \dot{\geq} \exp\left(-d_1^* d_0 \log N\right)$$

Next, if $d_1^* = 2N/d_0$ or $d_1^* \dot{<} N/d_0$ (is assumption 5 holds), we obtain the second (right) inequality

$$\exp\left(-d_1^* d_0 \log N\right) \dot{\geq} \exp\left(-2N \log N\right).$$

∎

## 11 VOLUME RATIO OF GLOBAL AND LOCAL MINIMA: PROOF OF THEOREM 10

**Theorem 24. (Theorem 10 restated)** *Given assumptions 1-3, we set* $\delta \doteq \sqrt{\frac{8}{\pi}} d_0^{-1/2} + 2d_0^{1/2}\sqrt{\log d_0}/N$. *Then, with probability* $1 - \delta$*, the angular volume of sub-optimal DLMs, with* MCE $> \epsilon > 0$*, is exponentially vanishing in N, in comparison to the angular volume of global minima with* MCE $= 0$

$$\frac{\mathcal{V}\left(\mathcal{L}_\epsilon\left(\mathbf{X}, \mathbf{y}\right)\right)}{\mathcal{V}\left(\mathcal{G}\left(\mathbf{X}, \mathbf{y}\right)\right)} \dot{\leq} \exp\left(-\gamma_\epsilon N^{3/4} [d_1 d_0]^{1/4}\right) \dot{\leq} \exp\left(-\gamma_\epsilon N \log N\right).$$

To prove this theorem we first calculate the expectation of the angular volume ratio given the $\mathbf{X}$-event that the bound in Theorem 9 holds (given assumptions 1-3), *i.e.*, $\mathcal{V}\left(\mathcal{G}\left(\mathbf{X}, \mathbf{y}\right)\right) \dot{\geq} \exp\left(-2N \log N\right)$. Denoting this event[6] as $\mathcal{M}$, we find:

$$\mathbb{E}_{\mathbf{X} \sim \mathcal{N}}\left[\frac{\mathcal{V}\left(\mathcal{L}_\epsilon\left(\mathbf{X}, \mathbf{y}\right)\right)}{\mathcal{V}\left(\mathcal{G}\left(\mathbf{X}, \mathbf{y}\right)\right)} | \mathcal{M}\right] \overset{(1)}{\dot{\leq}} \frac{\mathbb{E}_{\mathbf{X} \sim \mathcal{N}}\left[\mathcal{V}\left(\mathcal{L}_\epsilon\left(\mathbf{X}, \mathbf{y}\right)\right) | \mathcal{M}\right]}{\exp\left(-2N \log N\right)} \overset{(2)}{\dot{\leq}}$$

$$\frac{\mathbb{E}_{\mathbf{X} \sim \mathcal{N}}\left[\mathcal{V}\left(\mathcal{L}_\epsilon\left(\mathbf{X}, \mathbf{y}\right)\right)\right]}{\mathbb{P}_{\mathbf{X} \sim \mathcal{N}}\left(\mathcal{M}\right)\exp\left(-2N \log N\right)} \overset{(3)}{\dot{\leq}} \frac{\exp\left(-\gamma_\epsilon N^{3/4} [d_1 d_0]^{1/4}\right)}{\mathbb{P}_{\mathbf{X} \sim \mathcal{N}}\left(\mathcal{M}\right)\exp\left(-2N \log N\right)} \overset{(4)}{\dot{\leq}}$$

$$\frac{\exp\left(-\gamma_\epsilon N^{3/4} [d_1 d_0]^{1/4}\right)}{\exp\left(-2N \log N\right)} \overset{(5)}{\dot{\leq}} \exp\left(-\gamma_\epsilon N^{3/4} [d_1 d_0]^{1/4}\right) \tag{11.1}$$

where

1. We apply Theorem 9.
2. We use the following fact

---

[6]This event was previously denoted as $\mathbf{X} \in \mathcal{M}^\alpha\left(\mathbf{W}^*\right)$ in the proof of Theorem 9, but this is not important for this proof, so we simplified the notation.

**Fact 25.** *For any variable $X \geq 0$ and event $\mathcal{A}$ (where $\bar{\mathcal{A}}$ is its complement)*

$$\mathbb{E}\left[X\right] = \mathbb{E}\left[X|\mathcal{A}\right]\mathbb{P}\left(\mathcal{A}\right) + \mathbb{E}\left[X|\bar{\mathcal{A}}\right]\left(1 - \mathbb{P}\left(\mathcal{A}\right)\right) \geq \mathbb{E}\left[X|\mathcal{A}\right]\mathbb{P}\left(\mathcal{A}\right)$$

3. We apply Theorem 6.
4. We apply Theorem 9.
5. We use assumption 4, which implies $\gamma_\epsilon N^{3/4}\left[d_1 d_0\right]^{1/4} \dot{>} 2N\log N$.

For simplicity, in the reminder of the proof we denote

$$R\left(\mathbf{X}\right) \triangleq \frac{\mathcal{V}\left(\mathcal{L}_\epsilon\left(\mathbf{X}, \mathbf{y}\right)\right)}{\mathcal{V}\left(\mathcal{G}\left(\mathbf{X}, \mathbf{y}\right)\right)}.$$

From Markov inequality (Fact 11), since $R\left(\mathbf{X}\right) \geq 0$, we have $\forall \eta\left(N\right) > 0$:

$$\mathbb{P}_{\mathbf{X}\sim\mathcal{N}}\left[R\left(\mathbf{X}\right) \geq \eta\left(N\right)|\mathcal{M}\right] \leq \frac{\mathbb{E}_{\mathbf{X}\sim\mathcal{N}}\left[R\left(\mathbf{X}\right)|\mathcal{M}\right]}{\eta\left(N\right)} \qquad (11.2)$$

On the other hand, from fact 25, we have

$$1 - \mathbb{P}_{\mathbf{X}\sim\mathcal{N}}\left[R\left(\mathbf{X}\right) < \eta\left(N\right)|\mathcal{M}\right] \geq 1 - \frac{\mathbb{P}_{\mathbf{X}\sim\mathcal{N}}\left[R\left(\mathbf{X}\right) < \eta\left(N\right)\right]}{\mathbb{P}_{\mathbf{X}\sim\mathcal{N}}\left(\mathcal{M}\right)}. \qquad (11.3)$$

Combining Eqs. (11.2)-(11.3) we obtain

$$\frac{\mathbb{E}_{\mathbf{X}\sim\mathcal{N}}\left[R\left(\mathbf{X}\right)|\mathcal{M}\right]}{\eta\left(N\right)} \geq 1 - \frac{\mathbb{P}_{\mathbf{X}\sim\mathcal{N}}\left[R\left(\mathbf{X}\right) < \eta\left(N\right)\right]}{\mathbb{P}_{\mathbf{X}\sim\mathcal{N}}\left(\mathcal{M}\right)},$$

and so

$$\mathbb{P}_{\mathbf{X}\sim\mathcal{N}}\left(\mathcal{M}\right) - \mathbb{P}_{\mathbf{X}\sim\mathcal{N}}\left(\mathcal{M}\right)\frac{\mathbb{E}_{\mathbf{X}\sim\mathcal{N}}\left[R\left(\mathbf{X}\right)|\mathcal{M}\right]}{\eta\left(N\right)} \leq \mathbb{P}_{\mathbf{X}\sim\mathcal{N}}\left[R\left(\mathbf{X}\right) < \eta\left(N\right)\right].$$

We choose

$$\eta\left(N\right) = N\mathbb{P}_{\mathbf{X}\sim\mathcal{N}}\left(\mathcal{M}\right)\mathbb{E}_{\mathbf{X}\sim\mathcal{N}}\left[R\left(\mathbf{X}\right)|\mathcal{M}\right] \doteq \exp\left(-\gamma_\epsilon N^{3/4}\left[d_1 d_0\right]^{1/4}\right)$$

so that

$$\mathbb{P}_{\mathbf{X}\sim\mathcal{N}}\left(\mathcal{M}\right) - \frac{1}{N} \leq \mathbb{P}_{\mathbf{X}\sim\mathcal{N}}\left[R\left(\mathbf{X}\right) \dot{\leq} \exp\left(-\gamma_\epsilon N^{3/4}\left[d_1 d_0\right]^{1/4}\right)\right].$$

Then, from Theorem 9 we have

$$1 - \mathbb{P}_{\mathbf{X}\sim\mathcal{N}}\left(\mathcal{M}\right)\dot{\leq}\sqrt{\frac{8}{\pi}}d_0^{-1/2} + \frac{2d_0^{1/2}\sqrt{\log d_0}}{N}. \qquad (11.4)$$

so we obtain the first (left) inequality in the Theorem (10)

$$\sqrt{\frac{8}{\pi}}d_0^{-1/2} + \frac{2d_0^{1/2}\sqrt{\log d_0}}{N}\dot{\geq}1 - \mathbb{P}_{\mathbf{X}\sim\mathcal{N}}\left[\frac{\mathcal{V}\left(\mathcal{L}_\epsilon\left(\mathbf{X}, \mathbf{y}\right)\right)}{\mathcal{V}\left(\mathcal{G}\left(\mathbf{X}, \mathbf{y}\right)\right)}\dot{\leq}\exp\left(-\gamma_\epsilon N^{3/4}\left[d_1 d_0\right]^{1/4}\right)\right].$$

Lastly, we note that assumption 4 implies $\gamma_\epsilon N^{3/4}\left[d_1 d_0\right]^{1/4}\dot{>}N\log N$, which proves the second (right) inequality of the theorem.

$\blacksquare$

# Part II

# Proofs of technical results

In this part we prove the technical results used in part I.

## 12 UPPER BOUNDING THE ANGULAR VOLUME OF SUB-OPTIMAL DIFFERENTIABLE LOCAL MINIMA: PROOFS OF LEMMAS USED IN SECTION 8

### 12.1 PROOF OF LEMMA 14

In this section we will prove Lemma 14 in subsection 12.3.3. Recall the following definition

**Definition 26.** Let

$$\mathbf{A} = [\boldsymbol{a}_1, \dots, \boldsymbol{a}_N] \; ; \; \mathbf{X} = [\mathbf{x}_1, \dots, \mathbf{x}_N] \, ,$$

where $\mathbf{X} \in \mathbb{R}^{d_0 \times N}$ and $\mathbf{A} \in \mathbb{R}^{d_1 \times N}$. The Khatari-Rao product between the two matrices is defined as

$$
\begin{aligned}
\mathbf{A} \circ \mathbf{X} \quad &\triangleq \quad [\boldsymbol{a}_1 \otimes \mathbf{x}_1, \boldsymbol{a}_2 \otimes \mathbf{x}_2, \dots \boldsymbol{a}_N \otimes \mathbf{x}_N] \\
&= \quad \begin{pmatrix} a_{11}\mathbf{x}_1 & a_{12}\mathbf{x}_2 & \dots \\ a_{21}\mathbf{x}_1 & a_{22}\mathbf{x}_2 & \ddots \\ \vdots & \ddots & \ddots \end{pmatrix} .
\end{aligned}
\tag{12.1}
$$

**Lemma 27.** **(Lemma 14 restated)** *Let* $\mathbf{X} \in \mathbb{R}^{d_0 \times N}$, $\mathbf{A} \in \{\rho, 1\}^{d_1 \times N}$, $S \subset [N]$ *and* $d_0 d_1 \geq N$. *Then, simultaneously for every possible* $\mathbf{A}$ *and* $S$ *such that*

$$|S| \leq \operatorname{rank}(\mathbf{A}_S) \, d_0 \, ,$$

*we have that,* $\mathbf{X}$*-a.e.,* $\nexists \mathbf{v} \in \mathbb{R}^N$ *such that* $v_n \neq 0 \; \forall n \in S$ *and* $(\mathbf{A} \circ \mathbf{X}) \mathbf{v} = 0$ .

*Proof.* We examine specific $\mathbf{A} \in \{\rho, 1\}^{d_1 \times N}$ and $S \subset [N]$, and such that $|S| \leq d_S d_0$, where we defined $d_S \triangleq \operatorname{rank}(\mathbf{A}_S)$. We assume that $d_S \geq 1$, since otherwise the proof is trivial. Also, we assume by contradiction that $\exists \mathbf{v} \in \mathbb{R}^N$ such that $v_i \neq 0 \; \forall i \in S$ and $(\mathbf{A} \circ \mathbf{X}) \mathbf{v} = 0$ . Without loss of generality, assume that $S = \{1, 2, \dots, |S|\}$ and that $\boldsymbol{a}_1, \boldsymbol{a}_2, \dots, \boldsymbol{a}_{d_S}$ are linearly independent. Then

$$(\mathbf{A} \circ \mathbf{X}) \mathbf{v} = \sum_{n=1}^{|S|} v_n a_{k,n} \mathbf{x}_n = 0 \tag{12.2}$$

for every $1 \leq k \leq d_1$. From the definition of $S$ we must have $v_n \neq 0$ for every $1 \leq n \leq |S|$. Since $\boldsymbol{a}_1, \boldsymbol{a}_2, \dots, \boldsymbol{a}_{d_S}$ are linearly independent, the rows of $\mathbf{A}_{d_S} = [\boldsymbol{a}_1, \boldsymbol{a}_2, \dots, \boldsymbol{a}_{d_S}]$ span a $d_S$-dimensional space. Therefore, it is possible to find a matrix $\mathbf{R}$ such that $\mathbf{R}\mathbf{A}_{d_S} = [\mathbf{I}_{d_S \times d_S}, 0_{d_S \times (d_1 - d_S)}]^\top$, where $0_{i \times j}$ is the all zeros matrix with $i$ columns and $j$ rows. Consider now $\mathbf{A}_S \circ \mathbf{X}_S$, *i.e.*, the matrix composed of the columns of $\mathbf{A} \circ \mathbf{X}$ in $S$. Applying $\mathbf{R}' = \mathbf{R} \otimes \mathbf{I}_{d_0}$ to $\mathbf{A}_S \circ \mathbf{X}_S$, turns (12.2) into $d_0 d_S$ equations in the variables $v_1, \dots, v_{|S|}$, of the form

$$v_k \mathbf{x}_k + \sum_{n=d_S+1}^{|S|} v_n \tilde{a}_{k,n} \mathbf{x}_n = 0 \tag{12.3}$$

for every $1 \leq k \leq d_S$. We prove by induction that for every $1 \leq d \leq d_S$, the first $d_0 d$ equations are linearly independent, except for a set of matrices $\mathbf{X}$ of measure 0. This will immediately imply $|S| > d_S d_0$, or else eq. 12.2 cannot be true for $\mathbf{v} \neq 0$. which will contradict our assumption, as required. The induction can be viewed as carrying out Gaussian elimination of the system of equations described by (12.3), where in each elimination step we characterize the set of matrices $\mathbf{X}$ that for which that step is impossible, and show it has measure 0.

For $d = 1$, the first $d_0$ equations read $v_1\mathbf{x}_1 + \sum_{n=d_S+1}^{|S|} v_n\tilde{a}_{1,n}\mathbf{x}_n = 0$, and since $v_1 \neq 0$, we must have $\mathbf{x}_1 \in \mathrm{Span}\left\{\tilde{a}_{1,d_S+1}\mathbf{x}_{d_S+1}, ..., \tilde{a}_{1,|S|}\mathbf{x}_{|S|}\right\}$. However, except for a set of measure 0 with respect to $\mathbf{x}_1$ (a linear subspace of $\mathbb{R}^{d_0}$ with dimension less than $d_0$), this can only happen if $\dim \mathrm{Span}\left\{\tilde{a}_{1,d_S+1}\mathbf{x}_{d_S+1}, ..., \tilde{a}_{1,|S|}\mathbf{x}_{|S|}\right\} = d_0$, which implies $|S| \geq d_S - 1 + d_0 > d_0$ and also that the first $d_0$ rows are linearly independent (since there are $d_0$ independent columns).

For a general $d$, we begin by performing Gaussian elimination on the first $(d-1)\,d_0$ equations, resulting in a new set of $r_d$ equations, such that every new equation contains one variable that appears in no other new equation. Let $C$ be the set of the indices (equivalently, columns) of these variables $r_d$ variables. From (12.3) it is clear none of the variables $v_d, v_{d+1}, ..., v_{d_S}$ appear in the first $(d-1)\,d_0$ equations, and therefore $C \subseteq S' = S \setminus \{d, d+1, ..., d_S\}$. By our induction assumptions, except for a set of measure 0, the first $(d-1)\,d_0$ are independent, which means that $|C| = r_d = (d-1)\,d_0$. We now extend the Gaussian elimination to the next $d_0$ equations, and eliminate all the variables in $C$ from them. The result of the elimination can be written down as,

$$v_d\mathbf{x}_d + \sum_{n \in S' \setminus C} v_n \left(\tilde{a}_{d,n}\mathbf{I}_{d_0} - \mathbf{Y}\right) \mathbf{x}_n = 0\,, \tag{12.4}$$

where $\mathbf{Y}$ is a square matrix of size $d_0$ whose coefficients depend only on $\{\tilde{a}_{k,n}\}_{n \in C, d > k \geq 1}$ and on $\{\mathbf{x}_n\}_{n \in C}$, and in particular do not depend on $\mathbf{x}_d$ and $\{\mathbf{x}_n\}_{n \in S' \setminus C}$.

Now set $\tilde{\mathbf{x}}_n = (\tilde{a}_{d,n}\mathbf{I}_{d_0} - \mathbf{Y})\mathbf{x}_n$ for $n \in S' \setminus C$. As in the case of $d = 1$, since $v_d \neq 0$, $\mathbf{x}_d \in \mathrm{Span}\{\tilde{\mathbf{x}}_n\}_{n \in S' \setminus C}$. Therefore, for all values of $\mathbf{x}_d \in \mathbb{R}^{d_0}$ but a set of measure zero (linear subspace of with dimension less than $d_0$), we must have $\dim \mathrm{Span}\{\tilde{\mathbf{x}}_n\}_{n \in S' \setminus C} = d_0$. From the independence of $\{\tilde{\mathbf{x}}_n\}_{n \in S' \setminus C}$ on $\mathbf{x}_d$ it follows that $\dim \mathrm{Span}\{\tilde{\mathbf{x}}_n\}_{n \in S' \setminus C} = d_0$ holds a.e. with respect to the Lebesgue measure over $\mathbf{x}$.

Whenever $\dim \mathrm{Span}\{\tilde{\mathbf{x}}_n\}_{n \in S' \setminus C} = d_0$ we must have $|S' \setminus C| \geq d_0$ and therefore

$$|S| > |S'| = |C| + |S' \setminus C| \geq (d-1)\,d_0 + d_0 = d_0 d\,. \tag{12.5}$$

Moreover, $\dim \mathrm{Span}\{\tilde{\mathbf{x}}_n\}_{n \in S' \setminus C} = d_0$ implies that the $d_0$ equations $v_d\mathbf{x}_d + \sum_{n \in S' \setminus C} v_n\tilde{\mathbf{x}}_n = 0$ are independent. Thus, we may perform another step of Gaussian elimination on these $d_0$ equations, forming $d_0$ new equations each with a variable unique to it. Denoting by $C'$ the set of these $d_0$ variables, it is seen from (12.4) that $C' \subseteq (S' \cup \{d\}) \setminus C$ and in particular $C'$ is disjoint from $C$. Thus, considering the first $(d-1)\,d_0$ equations together with the new $d_0$ equations, we see that there is a set $C \cup C'$ of $d_0 d$ variables, such that each variable in $C \cup C'$ appears only in one of the $d_0 d$ equations, and each of the $d_0 d$ contains only a single variable in $C \cup C'$. This means that the first $d_0 d$ must be linearly independent for all values of $\mathbf{X}$ except for a set of Lebesgue measure zero, completing the induction.

Thus, we have proven, that for some $\mathbf{A} \in \{\rho, 1\}^{d_1 \times N}$ and $S \subset [N]$ such that $|S| \leq \mathrm{rank}\,(\mathbf{A}_S)\,d_0$ the event

$$\mathcal{E}\,(\mathbf{A}, S) = \left\{\mathbf{X} \in \mathbb{R}^{d_0 \times N} | \exists \mathbf{v} \in \mathbb{R}^N : (\mathbf{A} \circ \mathbf{X})\,\mathbf{v} = 0 \,\mathrm{and}\, v_n \neq 0, \forall n \in S\right\}$$

has zero measure. The event discussed in the theorem is a union of these events:

$$\mathcal{E}_0 \triangleq \bigcup_{\mathbf{A} \in \{\rho, 1\}^{d_1 \times N}} \left[\bigcup_{S \subset [N]: |S| \leq \mathrm{rank}(\mathbf{A}_S)d_0} \mathcal{E}\,(\mathbf{A}, S)\right]\,,$$

and it also has zero measure, since it is a finite union of zero measure events. $\square$

For completeness we note the following corollary, which is not necessary for a our main results.

**Corollary 28.** *If $N \leq d_1 d_0$, then $\mathrm{rank}\,(\mathbf{A} \circ \mathbf{X}) = N$, $\mathbf{X}$-a.e., if and only if,*

$$\forall S \subseteq [N] : |S| \leq \mathrm{rank}\,(\mathbf{A}_S)\,d_0\,.$$

*Proof.* We define $d_S \triangleq \mathrm{rank}\,(\mathbf{A}_S)$ and $\mathbf{A} \circ \mathbf{X}$. The necessity of the condition $|S| \leq d_0 d_S$ holds for every $\mathbf{X}$, as can be seen from the following counting argument. Since the matrix $\mathbf{A}_S$ has rank $d_S$,

there exists an invertible row transformation matrix $\mathbf{R}$, such that $\mathbf{R}\mathbf{A}_S$ has only $d_S$ non-zero rows. Consider now $\mathbf{G}_S = \mathbf{A}_S \circ \mathbf{X}_S$, *i.e.*, the matrix composed of the columns of $\mathbf{G}$ in $S$. We have

$$\mathbf{G}'_S = (\mathbf{R}\mathbf{A}_S) \circ \mathbf{X}_S = \mathbf{R}' (\mathbf{A}_S \circ \mathbf{X}_S) = \mathbf{R}'\mathbf{G}_S \,, \tag{12.6}$$

where $\mathbf{R}' = \mathbf{R} \otimes \mathbf{I}_{d_0}$ is also an invertible row transformation matrix, which applies $\mathbf{R}$ separately on the $d_0$ sub-matrices of $\mathbf{G}_S$ that are constructed by taking one every $d_0$ rows. Since $\mathbf{G}'_S$ has at most $d_0 d_S$ non-zero rows, the rank of $\mathbf{G}_S$ cannot exceed $d_0 d_S$. Therefore, if $|S| > d_0 d_S$, $\mathbf{G}_S$ will not have full column rank, and hence neither will $\mathbf{G}$. To demonstrate sufficiency a.e., suppose $\mathbf{G}$ does not have full column rank. Let $S$ be the minimum set of columns of $\mathbf{G}$ which are linearly dependent. Since the columns of $\mathbf{G}_S$ are assumed linearly dependent there exists $\mathbf{v} \in \mathbb{R}^{|S|}$ such $\|\mathbf{v}\|_0 = |S|$ and $\mathbf{G}_S \mathbf{v} = 0$. Using Lemma 28 we complete the proof. $\qquad\square$

## 12.2 Proof of Lemma 15

In this section we will prove Lemma 15 in subsection 12.3.3. This proof relies on two rather basic results, which we first prove in subsections 12.2.1 and 12.2.2.

### 12.2.1 Number of dichotomies induced by a hyperplane

**Fact 29.** *A hyperplane $\mathbf{w} \in d_0$ can separate a given set of points $\mathbf{X} = \left[\mathbf{x}^{(1)}, \ldots, \mathbf{x}^{(N)}\right] \in \mathbb{R}^{d_0 \times N}$ into several different dichotomies, i.e., different results for* $\mathrm{sign}\left(\mathbf{w}^\top \mathbf{X}\right)$. *The number of dichotomies is upper bounded as follows:*

$$\sum_{\mathbf{h} \in \{-1,1\}^N} \mathcal{I}\left(\exists \mathbf{w} : \mathrm{sign}\left(\mathbf{w}^\top \mathbf{X}\right) = \mathbf{h}^\top\right) \leq 2 \sum_{k=0}^{d_0 - 1} \binom{N-1}{k} \leq 2N^{d_0} \,. \tag{12.7}$$

*Proof.* See (Cover, 1965, Theorem 1) for a proof of the left inequality as equality (the Schläfli Theorem) in the case that the columns of $\mathbf{X}$ are in "general position" (which holds $\mathbf{X}$-a.e, see definition in (Cover, 1965)) . If $\mathbf{X}$ is not in general position then this result becomes an upper bound, since some dichotomies might not be possible.

Next, we prove the right inequality. For $N = 1$ and $N = 2$ the inequality trivially holds. For $N \geq 3$, we have

$$2 \sum_{k=0}^{d_0 - 1} \binom{N-1}{k} \overset{(1)}{\leq} 2 \sum_{k=0}^{d_0 - 1} (N-1)^k \overset{(2)}{\leq} 2 \frac{(N-1)^{d_0} - 1}{N-2} \leq 2N^{d_0} \,.$$

where in (1) we used the bound $\binom{N}{k} \leq N^k$ , in (2) we used the sum of a geometric series. $\quad\square$

### 12.2.2 A basic probabilistic bound

**Lemma 30.** *Let $\mathbf{H} = \left[\mathbf{h}_1^\top, \ldots, \mathbf{h}_{d_1}^\top\right]^\top \in \{-1,1\}^{d_1 \times k}$ be a deterministic binary matrix, $\mathbf{W} = \left[\mathbf{w}_1^\top, \ldots, \mathbf{w}_{d_1}^\top\right]^\top \in \mathbb{R}^{d_1 \times d_0}$ be an independent standard random Gaussian matrix, and $\mathbf{X} \in \mathbb{R}^{d_0 \times k}$ be a random matrix with independent and identically distributed columns.*

$$\mathbb{P}\left(\mathrm{sign}\left(\mathbf{W}\mathbf{X}\right) = \mathbf{H}\right) \leq \binom{k}{\lfloor k/2 \rfloor} \mathbb{P}\left(\mathbf{W}\mathbf{X}_{\lfloor k/2 \rfloor} > 0\right) \,.$$

*Proof.* By direct calculation

$$\mathbb{P}\left(\text{sign}\left(\mathbf{WX}\right)=\mathbf{H}\right)=\mathbb{E}\left[\mathbb{P}\left(\text{sign}\left(\mathbf{WX}\right)=\mathbf{H}|\mathbf{X}\right)\right]\overset{(1)}{=}\mathbb{E}\left[\prod_{i=1}^{d_1}\mathbb{P}\left(\text{sign}\left(\mathbf{w}_i^\top\mathbf{X}\right)=\mathbf{h}_i^\top|\mathbf{X}\right)\right]$$

$$\overset{(2)}{\leq}\mathbb{E}\left[\prod_{i=1}^{d_1}\mathbb{P}\left(\mathbf{w}_i^\top\mathbf{X}_{\hat{S}(\mathbf{h}_i)}>0|\mathbf{X}\right)\right]\overset{(3)}{\leq}\mathbb{E}\left[\prod_{i=1}^{d_1}\mathbb{P}\left(\mathbf{w}_i^\top\mathbf{X}_{S_*}>0|\mathbf{X}\right)\right]$$

$$\overset{(4)}{=}\mathbb{E}\left[\mathbb{P}\left(\mathbf{WX}_{S_*}>0|\mathbf{X}\right)\right]\overset{(5)}{\leq}\mathbb{E}\left[\sum_{S\subset[k]:|S|=\lfloor k/2\rfloor}\mathbb{P}\left(\mathbf{WX}_S>0|\mathbf{X}\right)\right]$$

$$=\sum_{S\subset[k]:|S|=\lfloor k/2\rfloor}\mathbb{E}\left[\mathbb{P}\left(\mathbf{WX}_S>0|\mathbf{X}\right)\right]\overset{(6)}{=}\binom{k}{\lfloor k/2\rfloor}\mathbb{P}\left(\mathbf{WX}_{[\lfloor k/2\rfloor]}>0\right).$$

where

1. We used the independence of the $\mathbf{w}_i$.

2. We define $\hat{S}_\pm(\mathbf{h})\triangleq\left\{S\subset[k]:\pm\mathbf{h}_S^\top>0\right\}$ as the sets in which $\mathbf{h}$ is always positive/negative, and $\hat{S}(\mathbf{h})$ as the maximal set between these two. Note that $\mathbf{w}_i$ has a standard normal distribution which is symmetric to sign flips, so $\forall S:\mathbb{P}\left(\mathbf{w}_i^\top\mathbf{X}_S>0|\mathbf{X}\right)=\mathbb{P}\left(\mathbf{w}_i^\top\mathbf{X}_S<0|\mathbf{X}\right)$.

3. Note that $\left|\hat{S}(\mathbf{h})\right|\geq\lfloor k/2\rfloor$. Therefore, we define $S_*=\underset{S\subset[k]:|S|=\lfloor k/2\rfloor}{\text{argmax}}\mathbb{P}\left(\mathbf{w}_i^\top\mathbf{X}_S>0|\mathbf{X}\right)$.

4. We used the independence of the $\mathbf{w}_i$.

5. The maximum is a single term in the following sum of non-negative terms.

6. Taking the expectation over $\mathbf{X}$, since the columns of $\mathbf{X}$ are independent and identically distributed, the location of $S$ does not affect the probability. Therefore, we can set without loss of generality $S=[\lfloor k/2\rfloor]$.

$\square$

### 12.2.3 Main proof: Bound on the number of configurations for a binary matrix with certain rank

Recall the function $a\left(\cdot\right)$ from eq. (2.1):

$$a\left(u\right)\triangleq\begin{cases}1&,\text{if},u>0\\\rho&,\text{if}\,u<0\end{cases}.$$

where $\rho\neq1$.

**Lemma 31. (Lemma 15 restated).** *Let* $\mathbf{X}\in\mathbb{R}^{d_0\times k}$ *be a random matrix with independent and identically distributed columns, and* $\mathbf{W}\in\mathbb{R}^{d_1\times d_0}$ *an independent standard random Gaussian matrix. Then, in the limit* $\min[k,d_0,d_1]\dot{>}r$,

$$\mathbb{P}\left(\text{rank}\left(a\left(\mathbf{WX}\right)\right)=r\right)\dot{\leq}2^{k+rd_0(\log d_1+\log k)+r^2}\mathbb{P}\left(\mathbf{WX}_{[\lfloor k/2\rfloor]}>0\right).$$

*Proof.* We denote $\mathbf{A}=a\left(\mathbf{WX}\right)\in\{\rho,1\}^{d_1\times k}$. For any such $\mathbf{A}$ for which $\text{rank}\left(\mathbf{A}\right)=r$, we have a collection of $r$ rows that span the remaining rows. There are $\binom{d_1}{r}$ possible locations for these $r$ spanning rows. In these rows there exist a collection of $r$ columns that span the remaining columns. There are $\binom{k}{r}$ possible locations for these $r$ spanning columns. At the intersection of the spanning

rows and columns, there exist a full rank sub-matrix $\mathbf{D}$. We denote $\tilde{\mathbf{A}}$ as the matrix $\mathbf{A}$ which rows and columns are permuted so that $\mathbf{D}$ is the lower right block

$$\tilde{\mathbf{A}} \triangleq \begin{pmatrix} \mathbf{Z} & \mathbf{B} \\ \mathbf{C} & \mathbf{D} \end{pmatrix} = a \begin{pmatrix} \mathbf{W}_1\mathbf{X}_1 & \mathbf{W}_1\mathbf{X}_2 \\ \mathbf{W}_2\mathbf{X}_1 & \mathbf{W}_2\mathbf{X}_2 \end{pmatrix}, \tag{12.8}$$

where $\mathbf{D}$ is an invertible $r \times r$ matrix, and we divided $\mathbf{X}$ and $\mathbf{W}$ to the corresponding block matrices

$$\mathbf{W} \triangleq \left[ \mathbf{W}_1^\top, \mathbf{W}_2^\top \right]^\top, \mathbf{X} \triangleq \left[ \mathbf{X}_1, \mathbf{X}_2 \right],$$

with $\mathbf{W}_2 \in \mathbb{R}^{r \times d_0}$ rows and $\mathbf{X}_2 \in \mathbb{R}^{d_0 \times r}$.

Since $\mathrm{rank}\left( \tilde{\mathbf{A}} \right) = r$, the first $d_1 - r$ rows are contained in the span of the last $r$ rows. Therefore, there exists a matrix $\mathbf{Q}$ such that $\mathbf{QC} = \mathbf{Z}$ and $\mathbf{QD} = \mathbf{B}$. Since $\mathbf{D}$ is invertible, this implies that $\mathbf{Q} = \mathbf{BD}^{-1}$ and therefore

$$\mathbf{Z} = \mathbf{BD}^{-1}\mathbf{C}, \tag{12.9}$$

*i.e.*, $\mathbf{B}, \mathbf{C}$ and $\mathbf{D}$ uniquely determine $\mathbf{Z}$.

Using the union bound over all possible permutations from $\mathbf{A}$ to $\tilde{\mathbf{A}}$, and eq. (12.9), we have

$$\mathbb{P}\left( \mathrm{rank}\left(\mathbf{A}\right) = r \right) \tag{12.10}$$

$$\leq \begin{pmatrix} d_1 \\ r \end{pmatrix}\begin{pmatrix} k \\ r \end{pmatrix} \mathbb{P}\left( \mathrm{rank}\left( \tilde{\mathbf{A}} \right) = r \right)$$

$$\leq \begin{pmatrix} d_1 \\ r \end{pmatrix}\begin{pmatrix} k \\ r \end{pmatrix} \mathbb{P}\left( \mathbf{Z} = \mathbf{BD}^{-1}\mathbf{C} \right)$$

$$= \begin{pmatrix} d_1 \\ r \end{pmatrix}\begin{pmatrix} k \\ r \end{pmatrix} \mathbb{P}\left( a\left(\mathbf{W}_1\mathbf{X}_2\right)\left[a\left(\mathbf{W}_2\mathbf{X}_2\right)\right]^{-1} a\left(\mathbf{W}_2\mathbf{X}_1\right) = a\left(\mathbf{W}_1\mathbf{X}_1\right) \right)$$

$$= \begin{pmatrix} d_1 \\ r \end{pmatrix}\begin{pmatrix} k \\ r \end{pmatrix} \sum_{\mathbf{H}\in\{-1,1\}^{(d_1-r)\times(k-r)}} \mathbb{P}\left( a\left(\mathbf{W}_1\mathbf{X}_2\right)\left[a\left(\mathbf{W}_2\mathbf{X}_2\right)\right]^{-1} a\left(\mathbf{W}_2\mathbf{X}_1\right) = a\left(\mathbf{H}\right) \,\middle|\, \mathrm{sign}\left(\mathbf{W}_1\mathbf{X}_1\right) = \mathbf{H} \right) \mathbb{P}\left( \mathrm{sign}\left(\mathbf{W}_1\mathbf{X}_1\right) = \mathbf{H} \right)$$

Using Lemma 30, we have

$$\mathbb{P}\left( \mathrm{sign}\left(\mathbf{W}_1\mathbf{X}_1\right) = \mathbf{H} \right) \leq \begin{pmatrix} k - r \\ \lfloor (k-r)/2 \rfloor \end{pmatrix} \mathbb{P}\left( \mathbf{W}_1\mathbf{X}_{[\lfloor (k-r)/2 \rfloor]} > 0 \right), \tag{12.11}$$

an upper bound which does not depend on $\mathbf{H}$. So all that remains is to compute the sum:

$$\sum_{\mathbf{H}\in\{-1,1\}^{(d_1-r)\times(k-r)}} \mathbb{P}\left( a\left(\mathbf{W}_1\mathbf{X}_2\right)\left[a\left(\mathbf{W}_2\mathbf{X}_2\right)\right]^{-1} a\left(\mathbf{W}_2\mathbf{X}_1\right) = a\left(\mathbf{H}\right) \,\middle|\, \mathrm{sign}\left(\mathbf{W}_1\mathbf{X}_1\right) = \mathbf{H} \right)$$

$$= \sum_{\mathbf{H}\in\{-1,1\}^{(d_1-r)\times(k-r)}} \mathbb{E}\left[ \mathbb{P}\left( a\left(\mathbf{W}_1\mathbf{X}_2\right)\left[a\left(\mathbf{W}_2\mathbf{X}_2\right)\right]^{-1} a\left(\mathbf{W}_2\mathbf{X}_1\right) = a\left(\mathbf{H}\right) \,\middle|\, \mathbf{W}_1, \mathbf{X}_1 \right) \,\middle|\, \mathrm{sign}\left(\mathbf{W}_1\mathbf{X}_1\right) = \mathbf{H} \right]$$

$$\overset{(1)}{\leq} \mathbb{E}\left[ \sum_{\mathbf{H}\in\{-1,1\}^{(d_1-r)\times(k-r)}} \mathcal{I}\left( \exists\left(\mathbf{W}_2, \mathbf{X}_2\right) : a\left(\mathbf{W}_1\mathbf{X}_2\right)\left[a\left(\mathbf{W}_2\mathbf{X}_2\right)\right]^{-1} a\left(\mathbf{W}_2\mathbf{X}_1\right) = a\left(\mathbf{H}\right) \right) \,\middle|\, \mathrm{sign}\left(\mathbf{W}_1\mathbf{X}_1\right) = \mathbf{H} \right] \tag{12.12}$$

$$\overset{(2)}{\leq} \mathbb{E}\left[ 2^{r^2}\left[ \sum_{\mathbf{H}\in\{-1,1\}^{(d_1-r)\times r}} \mathcal{I}\left( \exists\mathbf{X}_2 : \mathrm{sign}\left(\mathbf{W}_1\mathbf{X}_2\right) = \mathbf{H} \right) \right]\left[ \sum_{\mathbf{H}\in\{-1,1\}^{r\times(k-r)}} \mathcal{I}\left( \exists\mathbf{W}_2 : \mathrm{sign}\left(\mathbf{W}_2\mathbf{X}_1\right) = \mathbf{H} \right) \right] \,\middle|\, \mathrm{sign}\left(\mathbf{W}_1\mathbf{X}_1\right) = \mathbf{H} \right]$$

$$\leq \mathbb{E}\left[ 2^{r^2}\left[ \sum_{\mathbf{h}\in\{-1,1\}^{(d_1-r)}} \mathcal{I}\left( \exists\mathbf{x} : \mathrm{sign}\left(\mathbf{W}_1\mathbf{x}\right) = \mathbf{h} \right) \right]^r \left[ \sum_{\mathbf{h}\in\{-1,1\}^{(k-r)}} \mathcal{I}\left( \exists\mathbf{w} : \mathrm{sign}\left(\mathbf{w}^\top\mathbf{X}_1\right) = \mathbf{h}^\top \right) \right]^r \,\middle|\, \mathrm{sign}\left(\mathbf{W}_1\mathbf{X}_1\right) = \mathbf{H} \right]$$

$$\overset{(3)}{\leq} \mathbb{E}\left[ 2^{r^2} 2^{rd_0\log(d_1-r)+r} 2^{rd_0\log(k-r)+r} \,\middle|\, \mathrm{sign}\left(\mathbf{W}_1\mathbf{X}_1\right) = \mathbf{H} \right]$$

$$= 2^{rd_0[\log(d_1-r)+\log(k-r)]+r^2+2r}, \tag{12.13}$$

where

1. Given $(\mathbf{W}_1, \mathbf{X}_1)$, and eq. (12.8), the indicator function in eq. (12.12) is equal to zero only if $\mathbb{P}\left(a\left(\mathbf{W}_1\mathbf{X}_2\right)\left[a\left(\mathbf{W}_2\mathbf{X}_2\right)\right]^{-1}a\left(\mathbf{W}_2\mathbf{X}_1\right) = \mathbf{A}|\mathbf{W}_1, \mathbf{X}_1\right) = 0$, and one otherwise.

2. This sum counts the number of values of $\mathbf{H}$ consistent with $\mathbf{W}_1$ and $\mathbf{X}_1$. Conditioned on $(\mathbf{W}_1, \mathbf{X}_1)$, $\mathbf{D} = \left[a\left(\mathbf{W}_2\mathbf{X}_2\right)\right]^{-1}$, $\mathbf{B} = a\left(\mathbf{W}_1\mathbf{X}_2\right)$ and $\mathbf{C} = a\left(\mathbf{W}_2\mathbf{X}_1\right)$ can have multiple values, depending on $\mathbf{W}_2$ and $\mathbf{X}_2$. Also, any single value for $(\mathbf{D}, \mathbf{B}, \mathbf{C})$ results in a single value of $\mathbf{H}$. Therefore, the number of possible values of $\mathbf{H}$ in eq. (12.12) is upper bounded by the product of the number of possible values of $\mathbf{D}$, $\mathbf{B}$ and $\mathbf{C}$, which is product in the following equation.

3. The function $\sum_{\mathbf{h}\in\{-1,1\}^{(k-r)}} \mathcal{I}\left(\exists \mathbf{w} : \text{sign}\left(\mathbf{w}^\top \mathbf{X}_1\right) = \mathbf{h}^\top\right)$ counts the number of dichotomies that can be induced by the linear classifier $\mathbf{w}$ on $\mathbf{X}_1$. Using eq. (12.7) we can bound this number by $2\left(k-r\right)^{d_0}$. Similarly, the other sum can be bounded by $2\left(d_1 - r\right)^r$.

Combining eqs. (12.10), (12.11) and (12.13) we obtain

$$\mathbb{P}\left(\text{rank}\left(\mathbf{A}\right) = r\right) \leq$$
$$\binom{d_1}{r}\binom{k}{r}\binom{k-r}{\lfloor(k-r)/2\rfloor} 2^{rd_0[\log(d_1-r)+\log(k-r)]+r^2+2r}\mathbb{P}\left(\mathbf{W}_1\mathbf{X}_{[\lfloor(k-r)/2\rfloor]} > 0\right) .$$

Next, we take the log. To upper bound $\binom{N}{k}$, for small $k$ we use $\binom{N}{k} \leq N^k$, while for $k = N/2$, we use $\binom{N}{N/2} \leq 2^N$ . Thus, we obtain

$$\log \mathbb{P}\left(\text{rank}\left(\mathbf{A}\right) = r\right) \leq \left(rd_0\left(\log\left(d_1 - r\right) + \log\left(k - r\right)\right) + r^2 + 2r\right)\log 2 \tag{12.14}$$
$$+ r\log d_1 + r\log k + (k-r)\log 2 + \log \mathbb{P}\left(\mathbf{W}_1\mathbf{X}_{[\lfloor(k-r)/2\rfloor]} > 0\right) .$$

Recalling that $\mathbf{W}_1 \in \mathbb{R}^{(d_1-r)\times d_0}$ while $\mathbf{W} \in \mathbb{R}^{d_1\times d_0}$, we obtain from Jensen's inequality

$$\log \mathbb{P}\left(\mathbf{W}_1\mathbf{X}_{[\lfloor(k-r)/2\rfloor]} > 0\right) \leq \frac{\lfloor(k-r)/2\rfloor\lfloor d_1 - r\rfloor}{\lfloor k/2\rfloor\lfloor d_1\rfloor}\log \mathbb{P}\left(\mathbf{W}\mathbf{X}_{[\lfloor k/2\rfloor]} > 0\right) . \tag{12.15}$$

Taking the limit $\min\left[k, d_0, d_1\right] \dot{>} r$ on eqs. (12.14) and (12.15) we obtain

$$\mathbb{P}\left(\text{rank}\left(\mathbf{A}\right) = r\right) \dot{\leq} 2^{k+rd_0(\log d_1+\log k)+t^2}\mathbb{P}\left(\mathbf{W}\mathbf{X}_{[\lfloor k/2\rfloor]} > 0\right) .$$

$$\square$$

## 12.3 PROOF OF LEMMA 16

In this section we will prove Lemma 16 in subsection 12.3.3. This proof relies on more elementary results, which we first prove in subsections 12.3.1 and 12.3.2.

### 12.3.1 ORTHANT PROBABILITY OF A RANDOM GAUSSIAN VECTOR

Recall that $\phi\left(x\right)$ and $\Phi\left(x\right)$ are, respectively, the probability density function and cumulative distribution function for a scalar standard normal random variable.

**Definition 32.** We define the following functions $\forall x \geq 0$

$$g\left(x\right) \triangleq \frac{x\Phi\left(x\right)}{\phi\left(x\right)}, \tag{12.16}$$

$$\psi\left(x\right) \triangleq \frac{\left(g^{-1}\left(x\right)\right)^2}{2x} - \log\left(\Phi\left(g^{-1}\left(x\right)\right)\right), \tag{12.17}$$

where the inverse function $g^{-1}\left(x\right) : [0, \infty) \to [0, \infty)$ is well defined since $g\left(x\right)$ monotonically increase from 0 to $\infty$, for $x \geq 0$.

**Lemma 33.** *Let $\mathbf{z} \sim \mathcal{N}(0, \Sigma)$ be a random Gaussian vector in $\mathbb{R}^K$, with a covariance matrix $\Sigma_{ij} = (1 - \theta K^{-1}) \delta_{mn} + \theta K^{-1}$ where $K \gg \theta > 0$. Then, recalling $\psi(\theta)$ in eq. (12.17), we have*

$$\log \mathbb{P}(\forall i : z_i > 0) \leq -K\psi(\theta) + O(\log K).$$

*Proof.* Note that we can write $\mathbf{z} = \mathbf{u} + \eta$, where $\mathbf{u} \sim \mathcal{N}(0, (1 - \theta K^{-1})\mathbf{I}_K)$, and $\eta \sim \mathcal{N}(0, \theta K^{-1})$. Using this notation, we have

$$\mathbb{P}(\forall i : z_i > 0)$$

$$= \int_{-\infty}^{\infty} d\eta \left[ \prod_{i=1}^{K} \int_{-\infty}^{\infty} du_i \mathcal{I}\left( \sqrt{1 - \theta K^{-1}} u_i + \sqrt{\theta K^{-1}} \eta > 0 \right) \phi(u_i) \right] \phi(\eta)$$

$$= \int_{-\infty}^{\infty} d\eta \left[ \Phi\left( \sqrt{\frac{\theta K^{-1}}{1 - \theta K^{-1}}} \eta \right) \right]^{K} \phi(\eta)$$

$$\overset{(1)}{=} \sqrt{\frac{\theta}{2\pi(K - \theta)}} \int_{-\infty}^{\infty} d\xi \, [\Phi(\xi)]^{K} \exp\left( -\frac{(K - \theta)\xi^2}{2\theta} \right)$$

$$= \sqrt{\frac{\theta}{2\pi(K - \theta)}} \int_{-\infty}^{\infty} d\xi \exp\left( \frac{\xi^2}{2} \right) \exp\left[ K\left( \log \Phi(\xi) - \frac{\xi^2}{2\theta} \right) \right], \tag{12.18}$$

where in (1) we changed the variable of integration to $\xi = \sqrt{\theta/(K - \theta)}\eta$. We denote, for a fixed $\theta$,

$$q(\xi) \triangleq \log \Phi(\xi) - \frac{\xi^2}{2\theta} \tag{12.19}$$

$$h(\xi) \triangleq \sqrt{\frac{\theta}{2\pi(K - \theta)}} \exp\left( \frac{\xi^2}{2} \right) \tag{12.20}$$

and $\xi_0$ as its global maximum. Since $q$ is twice differentiable, we can use Laplace's method (*e.g.*, (Butler, 2007)) to simplify eq. (12.18)

$$\log \int_{-\infty}^{\infty} h(\xi) \exp(Kq(\xi)) \, d\xi = Kq(\xi_0) + O(\log K). \tag{12.21}$$

To find $\xi_0$, we differentiate $q(\xi)$ and equate to zero to obtain

$$q'(\xi) = \frac{\phi(\xi)}{\Phi(\xi)} - \frac{1}{\theta}\xi = 0. \tag{12.22}$$

which implies (recall eq. (12.16))

$$g(\xi) \triangleq \frac{\xi \Phi(\xi)}{\phi(\xi)} = \theta. \tag{12.23}$$

This is a monotonically increasing function from 0 to $\infty$ in the range $\xi \geq 0$. Its inverse function can also be defined in that range $g^{-1}(\theta) : [0, \infty] \to [0, \infty]$. This implies that this equation has only one solution, $\xi_0 = g^{-1}(\theta)$. Since $\lim_{\xi \to \infty} q(\xi) = -\infty$, this $\xi_0$ is indeed the global maximum of $q(\xi)$. Substituting this solution into $q(\xi)$, we get (recall eq. (12.17))

$$\forall \theta > 0 : q(\xi_0) = -\psi(\theta) = q(g^{-1}(\theta)) = \log(\Phi(g^{-1}(\theta))) - \frac{(g^{-1}(\theta))^2}{2\theta}. \tag{12.24}$$

Using eq. (12.18), (12.21) and (12.24) we obtain:

$$\log \mathbb{P}(\forall i : z_i > 0)$$

$$= \log \left[ \int_{-\infty}^{\infty} d\xi \exp\left( \frac{\xi^2}{2} \right) \exp\left[ K\left( \log \Phi(\xi) - \frac{\xi^2}{2\theta} \right) \right] \right] + O(\log K)$$

$$= -K\psi(\theta) + O(\log K).$$

$\square$

Next, we generalize the previous Lemma to a general covariance matrix.

**Corollary 34.** *Let* $\mathbf{u} \sim \mathcal{N}(0, \mathbf{\Sigma})$ *be a random Gaussian vector in* $\mathbb{R}^K$ *for which* $\forall n : \Sigma_{nn} = 1$, *and* $\theta \geq K \max_{n,m: n \neq m} \Sigma_{nm} > 0$. *Then, again, for large* $K$

$$\log \mathbb{P}(\forall i : u_i > 0) \leq -K\psi(\theta) + O(\log K).$$

*Proof.* We define $\tilde{\mathbf{u}} \sim \mathcal{N}\left(0, \tilde{\mathbf{\Sigma}}\right)$, with $\tilde{\Sigma}_{mn} = \left(1 - \theta K^{-1}\right) \delta_{mn} + \theta K^{-1}$. Note that $\forall n : \Sigma_{nn} = \tilde{\Sigma}_{nn} = 1$ and $\forall m \neq n$: $\Sigma_{mn} \leq \tilde{\Sigma}_{mn}$. Therefore, from Slepian's Lemma (Slepian, 1962, Lemma 1),

$$\mathbb{P}(\forall n : \tilde{u}_n > 0) \geq \mathbb{P}(\forall n : u_n > 0).$$

Using Lemma 33 on $\tilde{\mathbf{u}}$ completes the proof. $\qquad\square$

### 12.3.2 MUTUAL COHERENCE BOUNDS

**Definition 35.** We define the mutual coherence of the columns of a matrix $\mathbf{A} = [\boldsymbol{a}_1, \cdots, \boldsymbol{a}_N] \in \mathbb{R}^{M \times N}$ as the maximal angle between different columns

$$\gamma(\mathbf{A}) \triangleq \max_{i,j : i \neq j} \frac{\left|\boldsymbol{a}_i^\top \boldsymbol{a}_j\right|}{\|\boldsymbol{a}_i\| \|\boldsymbol{a}_j\|}.$$

Note that $\gamma(\mathbf{A}) \leq 1$ and from (Welch, 1974), for $N \geq M$, $\gamma(\mathbf{A}) \geq \sqrt{\frac{N-M}{M(N-1)}}$.

**Lemma 36.** *Let* $\mathbf{A} = [\boldsymbol{a}_1, \cdots, \boldsymbol{a}_N] \in \mathbb{R}^{M \times N}$ *be a standard random Gaussian matrix, and* $\gamma(\mathbf{A})$ *is the mutual coherence of it columns (see definition 35). Then*

$$\mathbb{P}(\gamma(\mathbf{A}) > \epsilon) \leq 2N^2 \exp\left(-\frac{M\epsilon^2}{24}\right).$$

*Proof.* In this case, we have from (Chen & Peng, 2016, Appendix 1):

$$\mathbb{P}(\gamma(\mathbf{A}) > \epsilon) \leq N(N-1)\left[\exp\left(-\frac{Ma^2\epsilon^2}{4(1+\epsilon/2)}\right) + \exp\left(-\frac{M}{4}(1-a)^2\right)\right],$$

for any $a \in (0, 1)$. Setting $a = 1 - \epsilon/2$

$$\mathbb{P}(\gamma(\mathbf{A}) > \epsilon) \leq N(N-1)\left[\exp\left(-\frac{M(1-\epsilon/2)^2 \epsilon^2}{4(1+\epsilon/2)}\right) + \exp\left(-\frac{M}{16}\epsilon^2\right)\right]$$

$$\overset{(1)}{\leq} N(N-1)\left[\exp\left(-\frac{M\epsilon^2}{24}\right) + \exp\left(-\frac{M}{16}\epsilon^2\right)\right]$$

$$\leq 2N^2 \exp\left(-\frac{M\epsilon^2}{24}\right),$$

where in (1) we can assume that $\epsilon \leq 1$, since for $\epsilon \geq 1$, we have $\mathbb{P}(\gamma(\mathbf{A}) > \epsilon) = 0$ (recall $\gamma(\mathbf{A}) \leq 1$). $\qquad\square$

**Lemma 37.** *Let* $\mathbf{B} = [\mathbf{b}_1, \cdots, \mathbf{b}_L] \in \mathbb{R}^{M \times L}$ *be a standard random Gaussian matrix and mutual coherence* $\gamma$ *as in definition 35. Then,* $\forall \epsilon > 0$ *and* $\forall K \in [L]$:

$$\mathbb{P}\left(\min_{S \subset [N] : |S| = K} \gamma(\mathbf{B}_S) > \epsilon\right) \leq \exp\left[\left(2\log(2K) - \frac{M\epsilon^2}{24}\right)\left(\frac{L}{K} - 1\right)\right].$$

*Proof.* We upper bound this probability by partitioning the set of column vectors into $\lfloor L/K \rfloor$ subsets $S_i$ of size $|S_i| = K$ and require that in each subset the mutual coherence is lower bounded by $\epsilon$.

Since the columns are independent, we have

$$\mathbb{P}\left(\min_{S \subset [N]:|S|=K} \gamma\left(\mathbf{B}_S\right) > \epsilon\right)$$

$$\leq \prod_{i=1}^{\lfloor L/K \rfloor} \mathbb{P}\left(\forall S = \{1 + (i-1)K, 2 + (1-i)K, \ldots, iK\} : \gamma\left(\mathbf{B}_S\right) > \epsilon\right)$$

$$\overset{(1)}{\leq} \prod_{i=1}^{L/K-1} 2K^2 \exp\left(-\frac{M\epsilon^2}{24}\right)$$

$$\leq \exp\left[\left(2\log\left(2K\right) - \frac{M\epsilon^2}{24}\right)\left(\frac{L}{K} - 1\right)\right],$$

where in (1) we used the bound from Lemma 36. $\qquad\square$

### 12.3.3 MAIN PROOF: ORTHANT PROBABILITY OF A PRODUCT GAUSSIAN MATRICES

**Lemma 38. (Lemma 16 restated).** *Let* $\mathbf{C} = [\mathbf{c}_1, \cdots, \mathbf{c}_N]^\top \in \mathbb{R}^{N \times M}$ *and* $\mathbf{B} \in \mathbb{R}^{M \times L}$ *be two independent random Gaussian matrices. Without loss of generality, assume* $N \geq L$, *and denote* $\alpha \triangleq ML/N$. *Then, in the regime* $M \leq N$ *and in the limit* $\min[N, M, L] \dot{>} \alpha \dot{>} 1$, *we have*

$$\mathbb{P}\left(\mathbf{CB} > 0\right) \dot{\leq} \exp\left(-0.4N\alpha^{1/4}\right).$$

*Proof.* For some $\theta > 0$, and subset $S$ such that $|S| = K < L$, we have

$$\mathbb{P}\left(\mathbf{CB} > 0\right)$$
$$\leq \mathbb{P}\left(\mathbf{CB}_S > 0 | \gamma\left(\mathbf{B}_S\right) \leq \epsilon\right)\mathbb{P}\left(\gamma\left(\mathbf{B}_S\right) \leq \epsilon\right) + \mathbb{P}\left(\mathbf{CB}_S > 0 | \gamma\left(\mathbf{B}_S\right) > \epsilon\right)\mathbb{P}\left(\gamma\left(\mathbf{B}_S\right) > \epsilon\right)$$
$$\leq \mathbb{P}\left(\mathbf{CB}_S > 0 | \gamma\left(\mathbf{B}_S\right) \leq \epsilon\right) + \mathbb{P}\left(\gamma\left(\mathbf{B}_S\right) > \epsilon\right)$$
$$= \mathbb{E}\left[\left[\mathbb{P}\left(\mathbf{c}_1^\top \mathbf{B}_S > 0 | \mathbf{B}_S, \gamma\left(\mathbf{B}_S\right) \leq \epsilon\right)\right]^N | \gamma\left(\mathbf{B}_S\right) \leq \epsilon\right] + \mathbb{P}\left(\gamma\left(\mathbf{B}_S\right) > \epsilon\right),$$

where in the last equality we used the fact that the rows of $\mathbf{C}$ are independent and identically distributed.

We choose a specific subset

$$S^* = \text{argmin}_{S \subset [L]:|S|=K} \gamma\left(\mathbf{B}_S\right)$$

to minimize the second term and then upper bound it using Lemma 37 with $\theta = K\epsilon$; additionally, we apply Corollary 34 on the first term with the components of the vector $\mathbf{u}$ being

$$u_i = \left(\mathbf{B}_S^\top \mathbf{c}_1\right)_i / \sqrt{\left(\mathbf{B}_S^\top \mathbf{B}_S\right)_{ii}} \in \mathbb{R}^K,$$

which is a Gaussian random vector with mean zero and covariance $\mathbf{\Sigma}$ for which $\forall i : \Sigma_{ii} = 1$ and $\forall i \neq j : \Sigma_{ij} \leq \epsilon = \theta K^{-1}$. Thus, we obtain

$$\mathbb{P}\left(\mathbf{CB} > 0\right) \leq \exp\left(-NK\psi\left(\theta\right) + O\left(N \log K\right)\right) + \exp\left[\left(\log\left(2K\right)^2 - \frac{M\theta^2}{24K^2}\right)\left(\frac{L}{K} - 1\right)\right], \tag{12.25}$$

where we recall $\psi\left(\theta\right)$ is defined in eq. (12.17).

Next, we wish to select good values for $\theta$ and $K$, which minimize this bound for large $(M, N, L, K)$. Thus, keeping only the first order terms in each exponent (assuming $L \gg K \gg 1$), we aim to minimize the function as much as possible

$$f\left(K, \theta\right) \triangleq \exp\left(-NK\psi\left(\theta\right)\right) + \exp\left(-\frac{M\theta^2 L}{24K^3}\right). \tag{12.26}$$

Note that the first term is decreasing in $K$, while the second term increases. Therefore, for any $\theta$ the minimum of this function in $K$ would be approximately achieved when both terms are equal, *i.e.*,

$$NK\psi\left(\theta\right) = \frac{M\theta^2 L}{24K^3},$$

so we choose

$$K\left(\theta\right) = \left(\frac{\theta^2 ML}{24\psi\left(\theta\right)N}\right)^{1/4}. \tag{12.27}$$

Substituting $K\left(\theta\right)$ into $f\left(K,\theta\right)$ yields

$$f\left(K\left(\theta\right),\theta\right) = 2\exp\left(-N\left[\frac{\psi^3\left(\theta\right)\theta^2 ML}{24N}\right]^{1/4}\right).$$

To minimize this function in $\theta$, we need to maximize the function $\psi^3\left(\theta\right)\theta^2$ (which has a single maximum). Doing this numerically gives us

$$\theta_* \approx 23.25\,;\; \psi\left(\theta_*\right) \approx 0.1062;\; \psi^3\left(\theta_*\right)\theta_*^2 \approx 0.6478\,. \tag{12.28}$$

Substituting eqs. (12.27) and (12.28) into eq. (12.25), we obtain

$$\mathbb{P}\left(\mathbf{CB} > 0\right)$$

$$\leq \exp\left(-N\left[\frac{ML}{37.05N}\right]^{1/4} + O\left(N\log K\right)\right)$$

$$+ \exp\left[-N\left[\frac{ML}{37.05N}\right]^{1/4} + 2L\frac{\log K}{K} + \frac{M\theta^2}{24K^2} - \log\left(2K^2\right)\right]$$

$$\leq \exp\left(-N\left[\frac{ML}{37.05N}\right]^{1/4} + O\left(N\log\left(\frac{ML}{N}\right)\right)\right),$$

where in the last line we used $N \geq L, N \geq M$ and $\min\left[N, M, L\right] \dot{>} \alpha \dot{>} 1$. Taking the log, and denoting $\alpha \triangleq ML/N$, we thus obtain

$$\log\mathbb{P}\left(\mathbf{CB} > 0\right) \leq -0.4N\alpha^{1/4} + O\left(N\log\alpha\right),$$

Therefore, in the limit that $N \to \infty$ and $\alpha\left(N\right) \to \infty$, with $\alpha\left(N\right) \dot{<} N$, we have

$$\mathbb{P}\left(\mathbf{CB} > 0\right) \dot{\leq} \exp\left(-0.4N\alpha^{1/4}\right).$$

$\square$

## 13 LOWER BOUNDING THE ANGULAR VOLUME OF GLOBAL MINIMA: PROOF OF LEMMAS USED IN SECTION 10

### 13.1 ANGLES BETWEEN RANDOM GAUSSIAN VECTORS

To prove the results in the next appendix sections, we will rely on the following basic Lemma.

**Lemma 39.** *For any vector* $\mathbf{y}$ *and* $\mathbf{x} \sim \mathcal{N}\left(0, \mathbf{I}_{d_0}\right)$*, we have*

$$\mathbb{P}\left(\left|\frac{\mathbf{x}^\top\mathbf{y}}{\|\mathbf{x}\|\,\|\mathbf{y}\|}\right| > \cos\left(\epsilon\right)\right) \geq \frac{2\sin\left(\epsilon\right)^{d_0-1}}{\left(d_0-1\right)B\left(\frac{1}{2}, \frac{d_0-1}{2}\right)} \tag{13.1}$$

$$\mathbb{P}\left(\left|\frac{\mathbf{x}^\top\mathbf{y}}{\|\mathbf{x}\|\,\|\mathbf{y}\|}\right| < u\right) \leq \frac{2u}{B\left(\frac{1}{2}, \frac{d_0-1}{2}\right)}, \tag{13.2}$$

*where we recall that* $B\left(x, y\right)$ *is the beta function.*

*Proof.* Since $\mathcal{N}\left(0, \mathbf{I}_{d_0}\right)$ is spherically symmetric, we can set $\mathbf{y} = \left[1, 0\ldots, 0\right]^\top$, without loss of generality. Therefore,

$$\left|\frac{\mathbf{x}^\top\mathbf{y}}{\|\mathbf{x}\|\,\|\mathbf{y}\|}\right|^2 = \frac{x_1^2}{x_1^2 + \sum_{i=2}^{d_0} x_i^2} \sim \mathcal{B}\left(\frac{1}{2}, \frac{d_0-1}{2}\right),$$

the Beta distribution, since $x_1^2 \sim \chi^2(1)$ and $\sum_{i=2}^{d_0} x_i^2 \sim \chi^2(d_0 - 1)$ are independent chi-square random variables.

Suppose $Z \sim \mathcal{B}(\alpha, \beta)$, $\alpha \in (0, 1)$, and $\beta > 1$.

$$\mathbb{P}(Z > u) = \frac{\int_u^1 x^{\alpha-1}(1-x)^{\beta-1}\, dx}{B(\alpha, \beta)} \geq \frac{\int_u^1 1^{\alpha-1}(1-x)^{\beta-1}\, dx}{B(\alpha, \beta)} = \frac{\int_0^{1-u} x^{\beta-1} dx}{B(\alpha, \beta)} = \frac{(1-u)^\beta}{\beta B(\alpha, \beta)}.$$

Therefore, for $\epsilon > 0$,

$$\mathbb{P}\left(\left|\frac{\mathbf{x}^\top \mathbf{y}}{\|\mathbf{x}\| \|\mathbf{y}\|}\right|^2 > \cos^2(\epsilon)\right) \geq \frac{2\left(1 - \cos^2(\epsilon)\right)^{\frac{d_0-1}{2}}}{(d_0 - 1) B\left(\frac{1}{2}, \frac{d_0-1}{2}\right)} = \frac{2\sin(\epsilon)^{d_0-1}}{(d_0 - 1) B\left(\frac{1}{2}, \frac{d_0-1}{2}\right)},$$

which proves eq. (13.1).

Similarly, for $\alpha \in (0, 1)$ and $\beta > 1$

$$\mathbb{P}(Z < u) = \frac{\int_0^u x^{\alpha-1}(1-x)^{\beta-1}\, dx}{B(\alpha, \beta)} \leq \frac{\int_0^u x^{\alpha-1} 1^{\beta-1} dx}{B(\alpha, \beta)} = \frac{u^\alpha}{\alpha B(\alpha, \beta)}.$$

Therefore, for $\epsilon > 0$,

$$\mathbb{P}\left(\left|\frac{\mathbf{x}^\top \mathbf{y}}{\|\mathbf{x}\| \|\mathbf{y}\|}\right|^2 < u^2\right) \leq \frac{2u}{B\left(\frac{1}{2}, \frac{d_0-1}{2}\right)},$$

which proves eq. (13.2). □

### 13.2 PROOF OF LEMMA 21:

Given three matrices: datapoints, $\mathbf{X} = \left[\mathbf{x}^{(1)}, \dots, \mathbf{x}^{(N)}\right] \in \mathbb{R}^{d_0 \times N}$, weights $\mathbf{W} = \left[\mathbf{w}_1^\top, \dots, \mathbf{w}_{d_1}^\top\right]^\top \in \mathbb{R}^{d_1 \times d_0}$, and target weights $\mathbf{W}^* = \left[\mathbf{w}_1^{*\top}, \dots, \mathbf{w}_{d_1^*}^{*\top}\right]^\top \in \mathbb{R}^{d_1^* \times d_0}$, with $d_1^* \leq d_1$, we recall the following definitions:

$$\mathcal{M}^\alpha(\mathbf{W}^*) \triangleq \left\{\mathbf{X} \in \mathbb{R}^{d_0 \times N} | \forall i, n : \left|\frac{\mathbf{x}^{(n)\top} \mathbf{w}_i^*}{\|\mathbf{x}^{(n)}\| \|\mathbf{w}_i^*\|}\right| > \sin\alpha\right\} \tag{13.3}$$

and

$$\tilde{\mathcal{G}}(\mathbf{X}, \mathbf{W}^*) \triangleq \left\{\mathbf{W} \in \mathbb{R}^{d_1 \times d_0} | \forall i \leq d_1^* : \ \mathrm{sign}\left(\mathbf{w}_i^\top \mathbf{X}\right) = \mathrm{sign}\left(\mathbf{w}_i^{*\top} \mathbf{X}\right)\right\}. \tag{13.4}$$

Using these definitions, in this section we prove the following Lemma.

**Lemma 40. (Lemma 21 restated).** *For any $\alpha$, if $\mathbf{W}^*$ is independent from $\mathbf{W}$ then, in the limit $N \to \infty$, $\forall \mathbf{X} \in \mathcal{M}^\alpha(\mathbf{W}^*)$ with $\log \sin \alpha \dot{>} d_0^{-1} \log d_0$*

$$\mathbb{P}_{\mathbf{W} \sim \mathcal{N}}\left(\mathbf{W} \in \tilde{\mathcal{G}}(\mathbf{X}, \mathbf{W}^*)\right) \dot{\geq} \exp\left(d_0 d_1^* \log \sin \alpha\right).$$

*Proof.* To lower bound $\mathbb{P}_{\mathbf{W} \sim \mathcal{N}}\left(\mathbf{W} \in \tilde{\mathcal{G}}(\mathbf{X}, \mathbf{W}^*)\right) \forall \mathbf{X} \in \mathcal{M}^\alpha(\mathbf{W}^*)$, we define the event that all weight hyperplanes (with normals $\mathbf{w}_i$) have an angle of at least $\alpha$ from the corresponding target hyperplanes (with normals $\mathbf{w}_i^*$).

$$\tilde{\mathcal{G}}_i^\alpha(\mathbf{W}^*) = \left\{\mathbf{W} \in \mathbb{R}^{d_1 \times d_0} | \left|\frac{\mathbf{w}_i^\top \mathbf{w}_i^*}{\|\mathbf{w}_i\| \|\mathbf{w}_i^*\|}\right| < \cos(\alpha)\right\}.$$

In order that $\mathrm{sign}\left(\mathbf{w}_i^\top \mathbf{x}^{(n)}\right) \neq \mathrm{sign}\left(\mathbf{w}_1^{*\top} \mathbf{x}^{(n)}\right)$, $\mathbf{w}_i$ must be rotated in respect to $\mathbf{w}_i^*$ by an angle greater then the angular margin $\alpha$, which is the minimal the angle between $\mathbf{x}^{(n)}$ and the solution hyperplanes (with normals $\mathbf{w}_i^*$). Therefore, we have that, given $\mathbf{X} \in \mathcal{M}^\alpha(\mathbf{W}^*)$,

$$\forall \alpha : \bigcap_{i=1}^{d_1^*} \tilde{\mathcal{G}}_i^\alpha(\mathbf{W}^*) \subset \tilde{\mathcal{G}}(\mathbf{X}, \mathbf{W}^*). \tag{13.5}$$

And so, $\forall \mathbf{X} \in \mathcal{M}^\alpha \left( \mathbf{W}^* \right)$ :

$$\mathbb{P}_{\mathbf{W} \sim \mathcal{N}} \left( \mathbf{W} \in \tilde{\mathcal{G}} \left( \mathbf{X}, \mathbf{W}^* \right) \right) \overset{(1)}{\geq} \mathbb{P}_{\mathbf{W} \sim \mathcal{N}} \left( \mathbf{W} \in \bigcap_{i=1}^{d_1^*} \tilde{\mathcal{G}}_i^\alpha \left( \mathbf{W}^* \right) \right) \tag{13.6}$$

$$\overset{(2)}{=} \prod_{i=1}^{d_1^*} \mathbb{P}_{\mathbf{W} \sim \mathcal{N}} \left( \mathbf{W} \in \tilde{\mathcal{G}}_i^\alpha \left( \mathbf{W}^* \right) \right) \overset{(3)}{\geq} \left[ \frac{2 \sin \left( \alpha \right)^{d_0 - 1}}{\left( d_0 - 1 \right) B \left( \frac{1}{2}, \frac{d_0 - 1}{2} \right)} \right]^{d_1^*},$$

where in (1) we used eq. (13.5), in (2) we used the independence of $\{\mathbf{w}_i\}_{i=1}^{d_1^*}$ and in (3) we used eq. (13.1) from Lemma 39. Lastly, to simplify this equation we use the asymptotic expansion of the beta function $B \left( \frac{1}{2}, x \right) = \sqrt{\pi/x} + O \left( x^{-3/2} \right)$ for large $x$:

$$\log \mathbb{P}_{\mathbf{W} \sim \mathcal{N}} \left( \mathbf{W} \in \tilde{\mathcal{G}} \left( \mathbf{X}, \mathbf{W}^* \right) \right) \geq d_0 d_1^* \log \sin \alpha + O \left( d_1^* \log d_0 \right).$$

We obtain the Lemma in the limit $N \to \infty$ when $\log \sin \alpha \dot{>} d_0^{-1} \log d_0$. $\qquad \square$

### 13.3 Proof of Lemma 22:

**Lemma 41. (Lemma 22 restated).** *Let* $\mathbf{W}^* = \left[ \mathbf{w}_1^\top, \ldots, \mathbf{w}_{d_1^*}^\top \right]^\top \in \mathbb{R}^{d_1^* \times d_0}$ *a fixed matrix independent of* $\mathbf{X}$. *Then, in the limit* $N \to \infty$ *with* $d_1^* \dot{\leq} d_0 \dot{\leq} N$, *the probability of not having an angular margin* $\sin \alpha = 1/ \left( d_1^* d_0 N \right)$ *(eq. (13.3)) is upper bounded by*

$$\mathbb{P} \left( \mathbf{X} \notin \mathcal{M}^\alpha \left( \mathbf{W}^* \right) \right) \dot{\leq} \sqrt{\frac{2}{\pi}} d_0^{-1/2}$$

*Proof.* We define

$$\mathcal{M}_{n,i}^\alpha \left( \mathbf{W}^* \right) \triangleq \left\{ \mathbf{X} \in \mathbb{R}^{d_0 \times N} | \left| \frac{\mathbf{x}^{(n) \top} \mathbf{w}_i^*}{\left\| \mathbf{x}^{(n)} \right\| \left\| \mathbf{w}_i^* \right\|} \right| > \sin \left( \alpha \right) \right\},$$

and $\mathcal{M}_n^\alpha \left( \mathbf{W}^* \right) \triangleq \bigcap_{i=1}^{d_1^*} \mathcal{M}_{n,i}^\alpha \left( \mathbf{W}^* \right)$. Since $\mathcal{M} \left( \mathbf{W}^* \right) = \bigcap_{n=1}^N \mathcal{M}_n^\alpha \left( \mathbf{W}^* \right)$, we have

$$\mathbb{P} \left( \mathbf{X} \in \mathcal{M}^\alpha \left( \mathbf{W}^* \right) \right) \overset{(1)}{=} \prod_{n=1}^N \mathbb{P} \left( \mathbf{X} \in \mathcal{M}_n^\alpha \left( \mathbf{W}^* \right) \right) = \prod_{n=1}^N \left[ 1 - \mathbb{P} \left( \mathbf{X} \notin \mathcal{M}_n^\alpha \left( \mathbf{W}^* \right) \right) \right]$$

$$\overset{(2)}{\geq} \prod_{n=1}^N \left[ 1 - \sum_{i=1}^{d_1^*} \mathbb{P} \left( \mathbf{X} \notin \mathcal{M}_{n,i}^\alpha \left( \mathbf{W}^* \right) \right) \right] \overset{(3)}{\geq} \left[ 1 - d_1^* \frac{2 \sin \left( \alpha \right)}{B \left( \frac{1}{2}, \frac{d_0 - 1}{2} \right)} \right]^N,$$

where in (1) we used the independence of $\left\{ \mathbf{x}^{(n)} \right\}_{n=1}^N$, in (2) we use the union bound, and in (3) we use eq. (13.2) from Lemma 39. Taking the log and we using the asymptotic expansion of the beta function $B \left( \frac{1}{2}, x \right) = \sqrt{\pi/x} + O \left( x^{-3/2} \right)$ for large $x$, we get

$$\log \mathbb{P} \left( \mathbf{X} \in \mathcal{M}^\alpha \left( \mathbf{W}^* \right) \right) \geq N \log \left[ 1 - \sqrt{\frac{2}{\pi}} d_0 d_1^* \sin \alpha + O \left( d_1^* d_0^{-1/2} \sin \alpha \right) \right]$$

$$= -\sqrt{\frac{2}{\pi}} d_0^{-1/2} + O \left( d_0^{-3/2}/N + d_0^{-1} N^{-2} \right),$$

where in the last line we recalled $\sin \alpha = 1/N$. Recalling that $d_1^* \dot{\leq} d_0 \dot{\leq} N$, we find

$$\mathbb{P} \left( \mathbf{X} \notin \mathcal{M}^\alpha \left( \mathbf{W}^* \right) \right) \dot{\geq} 1 - \exp \left( -\sqrt{\frac{2}{\pi}} d_0^{-1/2} \right) \geq \sqrt{\frac{2}{\pi}} d_0^{-1/2}$$

$\qquad \square$

### 13.4 PROOF OF LEMMA 23:

**Lemma 42. (Lemma 23 restated).** *Let $\mathbf{X} \in \mathbb{R}^{d_0 \times N}$ be a standard random Gaussian matrix of datapoints. Then we can find, with probability 1, $(\mathbf{X}, \mathbf{y})$-dependent matrices $\mathbf{W}^*$ and $\mathbf{z}^*$ as in Theorem 8 (where $d_1^* \triangleq 4 \lceil N/(2d_0 - 2) \rceil$). Moreover, in the limit $N \to \infty$, where $N/d_0 \dot{\le} d_0 \dot{\le} N$, for any $\mathbf{y}$, we can bound the probability of not having an angular margin (eq. (13.3)) with $\sin \alpha = 1/(d_1^* d_0 N)$ by*

$$\mathbb{P}\left(\mathbf{X} \notin \mathcal{M}^\alpha\left(\mathbf{W}^*\right)\right) \dot{\le} \sqrt{\frac{8}{\pi}} d_0^{-1/2} + \frac{2 d_0^{1/2} \sqrt{\log d_0}}{N}$$

*Proof.* In this proof we heavily rely on the notation and results from the proof of in appendix section 9. Without loss of generality we assume $\mathcal{S}_1^+ = [d_0 - 1]$. Unfortunately, we can't use Lemma 41 – this proof is significantly more complicated since the constructed solution $\mathbf{W}^*$ depends on $\mathbf{X}$ (we keep this dependence implicit, for brevity). Similarly to the proof of Lemma 41, we define,

$$\mathcal{M}_{i,n}^\alpha\left(\mathbf{W}^*\right) \triangleq \left\{ \mathbf{X} \in \mathbb{R}^{d_0 \times N} \, | \, \left| \frac{\mathbf{x}^{(n)\top} \mathbf{w}_i^*}{\left\|\mathbf{x}^{(n)}\right\| \left\|\mathbf{w}_i^*\right\|} \right| > \sin\left(\alpha\right) \right\}$$

and $\mathcal{M}_i^\alpha\left(\mathbf{W}^*\right) \triangleq \bigcap_{n=1}^N \mathcal{M}_{i,n}^\alpha\left(\mathbf{W}^*\right)$, so $\mathcal{M}\left(\mathbf{W}^*\right) = \bigcap_{i=1}^{d_1^*} \mathcal{M}_i^\alpha\left(\mathbf{W}^*\right)$. We have

$$\mathbb{P}\left(\mathbf{X} \in \mathcal{M}^\alpha\left(\mathbf{W}^*\right)\right) = 1 - \mathbb{P}\left(\mathbf{X} \notin \mathcal{M}^\alpha\left(\mathbf{W}^*\right)\right) \overset{(1)}{\ge} 1 - \sum_{i=1}^{d_1} \mathbb{P}\left(\mathbf{X} \notin \mathcal{M}_i^\alpha\left(\mathbf{W}^*\right)\right)$$

$$\overset{(2)}{=} 1 - d_1^* \mathbb{P}\left(\mathbf{X} \notin \mathcal{M}_1^\alpha\left(\mathbf{W}^*\right)\right) = 1 - d_1^* \left(1 - \mathbb{P}\left(\mathbf{X} \in \mathcal{M}_1^\alpha\left(\mathbf{W}^*\right)\right)\right), \quad (13.7)$$

where in (1) we used the union bound, and in (2) we used the fact that, from symmetry, $\forall i : \mathbb{P}\left(\mathbf{X} \notin \mathcal{M}_i^\alpha\left(\mathbf{W}^*\right)\right) = \mathbb{P}\left(\mathbf{X} \notin \mathcal{M}_1^\alpha\left(\mathbf{W}^*\right)\right)$. Next, we examine the minimal angular margin in $\mathcal{M}_{1,n}^\alpha$: separately for $\forall n < d_0$ and $\forall n \ge d_0$. Recalling the construction of $\mathbf{W}$ in appendix section 9, we have, for $\forall n < d_0$:

$$\min_{i, n < d_0} \left| \frac{\mathbf{x}^{(n)\top} \mathbf{w}_i^*}{\left\|\mathbf{x}^{(n)}\right\| \left\|\mathbf{w}_i^*\right\|} \right| = \min_{n < d_0, \pm} \frac{\left|\left(\tilde{\mathbf{w}}_1 \pm \epsilon_2 \hat{\mathbf{w}}_1\right)^\top \mathbf{x}^{(n)}\right|}{\left\|\tilde{\mathbf{w}}_1 \pm \epsilon_2 \hat{\mathbf{w}}_1\right\| \left\|\mathbf{x}^{(n)}\right\|}$$

$$\overset{(1)}{=} \min_{n < d_0, \pm} \frac{\epsilon_2}{\left\|\tilde{\mathbf{w}}_1 \pm \epsilon_2 \hat{\mathbf{w}}_1\right\| \left\|\mathbf{x}^{(n)}\right\|} \overset{(2)}{=} \frac{\gamma \epsilon_1 / \sqrt{1 + \gamma^2 \epsilon_1^2}}{\left\|\hat{\mathbf{w}}_1\right\| \max_{n < d_0} \left\|\mathbf{x}^{(n)}\right\|}, \quad (13.8)$$

where in (1) we used $\forall n < d_0$: $\mathbf{x}^{(n)\top} \hat{\mathbf{w}}_1 = 1$ and $\mathbf{x}^{(n)\top} \tilde{\mathbf{w}}_1 = 0$, from the construction of $\tilde{\mathbf{w}}_1$ and $\hat{\mathbf{w}}_1$ (eqs. (9.2), (9.5), and (9.4)), and in (2) we used the fact that $\hat{\mathbf{w}}_1^\top \tilde{\mathbf{w}}_1 = 0$ from eq. (9.4) together with $\left\|\tilde{\mathbf{w}}_1\right\| = \left\|\hat{\mathbf{w}}_1\right\|$ from eq. (9.5), and $\epsilon_2 = \gamma \epsilon_1$ from eq. (9.7).

For $\forall n \ge d_0$ :

$$\min_{i, n \ge d_0} \left| \frac{\mathbf{x}^{(n)\top} \mathbf{w}_i^*}{\left\|\mathbf{x}^{(n)}\right\| \left\|\mathbf{w}_i^*\right\|} \right| = \min_{n \ge d_0, \pm} \frac{\left|\left(\tilde{\mathbf{w}}_1 \pm \epsilon_1 \hat{\mathbf{w}}_1\right)^\top \mathbf{x}^{(n)}\right|}{\left\|\tilde{\mathbf{w}}_1 \pm \epsilon_1 \hat{\mathbf{w}}_1\right\| \left\|\mathbf{x}^{(n)}\right\|} \ge \frac{\left(1 - \gamma \beta\right) \epsilon_1}{\gamma \beta \sqrt{1 + \epsilon_1^2}} \min_{n \ge d_0} \frac{\left|\hat{\mathbf{w}}_1^\top \mathbf{x}^{(n)}\right|}{\left\|\hat{\mathbf{w}}_1\right\| \left\|\mathbf{x}^{(n)}\right\|},$$
$$(13.9)$$

where we used the fact that $\forall n \ge d_0$ : $\epsilon_2 \left|\hat{\mathbf{w}}_1^\top \mathbf{x}^{(n)}\right| \le \gamma \beta \left|\tilde{\mathbf{w}}_1^\top \mathbf{x}^{(n)}\right|$, from eq. (9.7), and also that $\hat{\mathbf{w}}_1^\top \tilde{\mathbf{w}}_1 = 0$ from eq. (9.4).

We substitute eqs. (13.8) and (13.9) into $\mathbb{P}\left(\mathbf{X} \in \mathcal{M}_1^\alpha\left(\mathbf{W}^*\right)\right)$:
$\mathbb{P}\left(\mathbf{X} \in \mathcal{M}_1^\alpha\left(\mathbf{W}^*\right)\right)$

$$\ge \mathbb{P}\left( \frac{\gamma \epsilon_1 / \sqrt{1 + \gamma^2 \epsilon_1^2}}{\left\|\hat{\mathbf{w}}_1\right\| \max_{n < d_0} \left\|\mathbf{x}^{(n)}\right\|} > \sin \alpha, \frac{\left(1 - \gamma \beta\right) \epsilon_1}{\gamma \beta \sqrt{1 + \epsilon_1^2}} \min_{n \ge d_0} \frac{\left|\hat{\mathbf{w}}_1^\top \mathbf{x}^{(n)}\right|}{\left\|\hat{\mathbf{w}}_1\right\| \left\|\mathbf{x}^{(n)}\right\|} > \sin \alpha \right)$$

$$\overset{(1)}{\ge} \mathbb{P}\left( \frac{\gamma \kappa}{\left\|\hat{\mathbf{w}}_1\right\| \max_{n < d_0} \left\|\mathbf{x}^{(n)}\right\|} > \sin \alpha, \frac{\left(1 - \gamma \beta\right)}{\gamma \beta} \kappa \min_{n \ge d_0} \frac{x_1^{(n)}}{\left\|\mathbf{x}^{(n)}\right\|} > \sin \alpha, \frac{\epsilon_1}{\sqrt{1 + \epsilon_1^2}} > \kappa \right)$$
$$(13.10)$$

$$\overset{(2)}{\ge} \mathbb{P}\left( \frac{\gamma \kappa}{\eta \sin \alpha} > \left\|\hat{\mathbf{w}}_1\right\|, \eta > \max_{n < d_0} \left\|\mathbf{x}^{(n)}\right\| \right) \mathbb{P}\left( \frac{\left(1 - \gamma \beta\right)}{\gamma \beta} \kappa \min_{n \ge d_0} \frac{x_1^{(n)}}{\left\|\mathbf{x}^{(n)}\right\|} > \sin \alpha, \frac{\epsilon_1}{\sqrt{1 + \epsilon_1^2}} > \kappa \right),$$

where in (1) we rotate the axes so that $\hat{\mathbf{w}}_1 \propto [1, 0, 0 \ldots, 0]$ axes $\tilde{\mathbf{w}}_1 \propto [0, 1, 0, 0 \ldots, 0]$ – this is possible due to the spherical symmetry of $\mathbf{x}^{(n)}$, and the fact that $\hat{\mathbf{w}}_1$ and $\tilde{\mathbf{w}}_1$ are functions of $\mathbf{x}^{(n)}$ for $n < d_0$ (from eqs. (9.4) and (9.2)), and as such, they are independent from $\mathbf{x}^{(n)}$ for $n \geq d_0$, in (2) we use that fact that $\|\hat{\mathbf{w}}_1\|$ and $\max_{n<d_0} \left\|\mathbf{x}^{(n)}\right\|$ are functions of $\mathbf{x}^{(n)}$ for $n < d_0$, and as such, they are independent from $\mathbf{x}^{(n)}$ for $n \geq d_0$. Thus,

$$
\begin{aligned}
&\mathbb{P}\left(\mathbf{X} \in \mathcal{M}_1^\alpha\left(\mathbf{W}^*\right)\right) \\
&\geq \left(1 - \mathbb{P}\left(\frac{\gamma\kappa}{\eta\sin\alpha} \leq \|\hat{\mathbf{w}}_1\| \text{ or } \eta \leq \max_{n<d_0}\left\|\mathbf{x}^{(n)}\right\|\right)\right) \\
&\quad \cdot \left(1 - \mathbb{P}\left(\frac{(1-\gamma\beta)}{\gamma\beta}\kappa\min_{n\geq d_0}\frac{x_1^{(n)}}{\left\|\mathbf{x}^{(n)}\right\|} \leq \sin\alpha \text{ or } \frac{\epsilon_1}{\sqrt{1+\epsilon_1^2}} \leq \kappa\right)\right) \\
&\overset{(1)}{\geq} \left(1 - \mathbb{P}\left(\frac{\gamma\kappa}{\eta\sin\alpha} \leq \|\hat{\mathbf{w}}_1\|\right) - \mathbb{P}\left(\eta \leq \max_{n<d_0}\left\|\mathbf{x}^{(n)}\right\|\right)\right) \\
&\quad \cdot \left(1 - \mathbb{P}\left(\frac{(1-\gamma\beta)}{\gamma\beta}\kappa\min_{n\geq d_0}\frac{x_1^{(n)}}{\left\|\mathbf{x}^{(n)}\right\|} \leq \sin\alpha\right) - \mathbb{P}\left(\frac{\epsilon_1}{\sqrt{1+\epsilon_1^2}} \leq \kappa\right)\right) \\
&= \left(\mathbb{P}\left(\eta > \max_{n<d_0}\left\|\mathbf{x}^{(n)}\right\|\right) - \mathbb{P}\left(\frac{\gamma\kappa}{\eta\sin\alpha} \leq \|\hat{\mathbf{w}}_1\|\right)\right) \\
&\quad \cdot \left(\mathbb{P}\left(\frac{(1-\gamma\beta)}{\gamma\beta}\kappa\min_{n\geq d_0}\frac{x_1^{(n)}}{\left\|\mathbf{x}^{(n)}\right\|} > \sin\alpha\right) - \mathbb{P}\left(\frac{\epsilon_1}{\sqrt{1+\epsilon_1^2}} \leq \kappa\right)\right),
\end{aligned}
\tag{13.11}
$$

where in (1) we use the union bound on both probability terms.

All that remains is to calculate each remaining probability term in eq. (13.11). First, we have

$$
\begin{aligned}
\mathbb{P}\left(\frac{\epsilon_1}{\sqrt{1+\epsilon_1^2}} \leq \kappa\right) &= 1 - \mathbb{P}\left(\frac{\kappa}{\sqrt{1-\kappa^2}} < \epsilon_1\right) \\
&\overset{(1)}{=} 1 - \mathbb{P}\left(\min_{n\geq d_0}\frac{\left|\tilde{\mathbf{w}}_i^\top\mathbf{x}^{(n)}\right|}{\left|\hat{\mathbf{w}}_i^\top\mathbf{x}^{(n)}\right|} > \frac{\kappa}{\sqrt{1-\kappa^2}}\frac{1}{\beta}\right) \overset{(2)}{=} 1 - \mathbb{P}\left(\min_{n\geq d_0}\left|\frac{x_2^{(n)}}{x_1^{(n)}}\right| > \frac{\kappa}{\sqrt{1-\kappa^2}}\frac{1}{\beta}\right) \\
&\overset{(3)}{=} 1 - \left[\mathbb{P}\left(\left|\frac{x_2^{(1)}}{x_1^{(1)}}\right| > \frac{\kappa}{\sqrt{1-\kappa^2}}\frac{1}{\beta}\right)\right]^{N-d_0-1} \overset{(4)}{\leq} 1 - \left[1 - \frac{2}{\pi}\arctan\left(\frac{\kappa}{\sqrt{1-\kappa^2}}\frac{1}{\beta}\right)\right]^N,
\end{aligned}
\tag{13.12}
$$

where in (1) we used eq. (9.7), in (2) we recall that in eq. (13.10) we rotated the axes so that $\hat{\mathbf{w}}_1 \propto [1, 0, 0 \ldots, 0]$ axes $\tilde{\mathbf{w}}_1 \propto [0, 1, 0, 0 \ldots, 0]$, in (3) we used the independence of different $\mathbf{x}^{(n)}$, and in (4) we used the fact that the ratio of two independent Gaussian variables is distributed according to the symmetric Cauchy distribution, which has the cumulative distribution function $\mathbb{P}(X > x) = \frac{1}{2} - \frac{1}{\pi}\arctan(x)$, and therefore $\mathbb{P}(|X| > x) = 1 - \frac{2}{\pi}\arctan(x)$.

Second, we use eq. (13.2)

$$
\mathbb{P}\left(\min_{n\geq d_0}\frac{x_1^{(n)}}{\left\|\mathbf{x}^{(n)}\right\|} > \frac{\gamma\beta\sin\alpha}{(1-\gamma\beta)\kappa}\right) > \left[1 - \frac{2\gamma\beta\sin\alpha}{(1-\gamma\beta)\kappa B\left(\frac{1}{2}, \frac{d_0-1}{2}\right)}\right]^N.
\tag{13.13}
$$

Third, $\left\|\mathbf{x}^{(n)}\right\|^2$ is distributed according to the chi-square distribution of order $d_0$, so for $\eta^2 > d_0$,

$$
\mathbb{P}\left(\left\|\mathbf{x}^{(n)}\right\|^2 \geq \eta^2\right) \leq \left(\eta^2\exp\left(1-\eta^2/d_0\right)/d_0\right)^{d_0/2}.
$$

Therefore,

$$
\mathbb{P}\left(\max_{n<d_0}\left\|\mathbf{x}^{(n)}\right\|^2 < \eta^2\right) > \left[1 - \left(\eta^2\exp\left(1-\eta^2/d_0\right)/d_0\right)^{d_0/2}\right]^{d_0-1}.
\tag{13.14}
$$

Lastly, we bound $\|\tilde{\mathbf{w}}_1\| = \|\hat{\mathbf{w}}_1\|$ (from eq. (9.5)). From eq. (9.4), we have

$$\hat{\mathbf{w}}_1^\top \mathbf{X}_{[d_0-1]} = [1, \ldots, 1, 1] , \tag{13.15}$$

where $\mathbf{X}_{[d_0-1]}$ has a singular value decomposition

$$\mathbf{X}_{[d_0-1]} = \sum_{i=1}^{d_0} \sigma_i \mathbf{u}_i \mathbf{v}_i^\top ,$$

with $\sigma_i$ being the singular values, and $\mathbf{u}_i$ and $\mathbf{v}_i$ being the singular vectors. The singular values are ordered from smallest to largest, and $\sigma_1 = 0$ with $\mathbf{u}_1 = \tilde{\mathbf{w}}_1$, from eq. (9.2). With probability 1, the other $d_0 - 1$ singular value are non-zero: they are the square roots of the eigenvalues of the random matrix $\mathbf{X}_{[d_0-1]}^\top \mathbf{X}_{[d_0-1]} \in \mathbb{R}^{d_0-1 \times d_0-1}$. Taking the squared norm of eq. (13.15), we have

$$d_0 - 1 = \hat{\mathbf{w}}_1^\top \mathbf{X}_{[d_0-1]} \mathbf{X}_{[d_0-1]}^\top \hat{\mathbf{w}}_1 = \sum_{i=1}^{d_0} \sigma_i^2 \left( \mathbf{u}_i^\top \hat{\mathbf{w}}_1 \right)^2 \geq \sigma_2^2 \|\hat{\mathbf{w}}_1\|^2 , \tag{13.16}$$

where the last inequality stems from the fact that $\mathbf{u}_1^\top \hat{\mathbf{w}}_1 = \tilde{\mathbf{w}}_1^\top \hat{\mathbf{w}}_1 = 0$ (from eq. (9.4)), so the minimal possible value is attained when $\mathbf{u}_2^\top \hat{\mathbf{w}}_1 = \|\hat{\mathbf{w}}_1\|$. The minimal nonzero singular value, $\sigma_2$, can be bounded using the following result from (Rudelson & Vershynin, 2010, eq. (3.2))

$$\mathbb{P}\left( \min_{\mathbf{r} \in \mathbb{R}^{d_0}} \left\| \mathbf{X}_{[d_0]} \mathbf{r} \right\| \leq \eta d_0^{-1/2} \right) \leq \eta.$$

Since

$$\sigma_2 = \min_{\mathbf{r} \in \mathbb{R}^{d_0-1}} \left\| \mathbf{X}_{[d_0-1]} \mathbf{r} \right\| \geq \min_{\mathbf{r} \in \mathbb{R}^{d_0}} \left\| \mathbf{X}_{[d_0]} \mathbf{r} \right\|$$

we have,

$$\mathbb{P}\left( \sigma_2 < \eta d_0^{-1/2} \right) \leq \eta.$$

Combining this with eq. (13.16) we get

$$\mathbb{P}\left( \frac{\beta \kappa}{\eta \sin \alpha} < \|\mathbf{w}_1\| \right) \leq \frac{\eta d_0}{\beta \kappa} \sin \alpha. \tag{13.17}$$

Lastly, combining eqs. (13.12), (13.13), (13.14) and (13.17) into eqs. (13.7) and (13.11), we get, for $\eta^2 > d_0$,

$$\mathbb{P}\left( \mathbf{X} \in \mathcal{M}^\alpha \left( \mathbf{W}^* \right) \right)$$

$$\geq 1 - d_1^* \left( 1 - \left( \left[ 1 - \left( \eta^2 \exp\left( 1 - \eta^2/d_0 \right) /d_0 \right)^{d_0/2} \right]^{d_0-1} - \frac{\eta d_0}{\gamma \kappa} \sin \alpha \right) \right.$$

$$\left. \cdot \left( \left[ 1 - \frac{2\gamma\beta \sin \alpha}{(1 - \gamma\beta) \kappa B \left( \frac{1}{2}, \frac{d_0-1}{2} \right)} \right]^N - \left[ 1 - \frac{2}{\pi} \arctan\left( \frac{\kappa}{\sqrt{1 - \kappa^2}} \frac{1}{\beta} \right) \right]^N \right) \right)$$

$$\geq 1 - d_1^* \left( 1 - \left( \left[ 1 - \left( \log d_0 \exp\left( 1 - \log d_0 \right) \right)^{d_0/2} \right]^{d_0-1} - \frac{2 d_0^{3/2} \sqrt{\log d_0}}{d_1^* N} \right) \right.$$

$$\left. \left( \left[ 1 - \sqrt{\frac{8}{\pi}} \frac{1}{d_1^* d_0^{1/2} N} + O\left( \frac{1}{N d_1^* d_0^{3/2}} \right) \right]^N - 0.45^N \right) \right) ,$$

where in the last line we take $\beta = \gamma = \kappa = 1/\sqrt{2}$, $\eta = d_0^{1/2} \sqrt{\log d_0}$, $\sin \alpha = 1/ (d_1^* d_0 N)$. Using the asymptotic expansion of the beta function $B\left( \frac{1}{2}, x \right) = \sqrt{\pi/x} + O\left( x^{-3/2} \right)$ for large $x$, we obtain,

for $\sin \alpha = 1/(d_1^* d_0 N)$

$1 - \mathbb{P}(\mathbf{X} \in \mathcal{M}^\alpha(\mathbf{W}^*))$

$$\leq d_1^* \left( 1 - \left( \left[ 1 - \exp\left( -\frac{d_0}{2} \log\left( \frac{d_0}{e \log d_0} \right) \right) \right]^{d_0 - 1} - \frac{2 d_0^{1/2} \sqrt{\log d_0}}{d_1^* N} \right) \right.$$

$$\left. \cdot \left( \left[ 1 - \sqrt{\frac{8}{\pi}} \frac{1}{N d_1^* d_0^{1/2}} + O\left( \frac{1}{N d_1^* d_0^{3/2}} \right) \right]^N - 2^{-N} \right) \right)$$

$$= d_1^* \left( 1 - \left( 1 - \frac{2 d_0^{1/2} \sqrt{\log d_0}}{d_1^* N} + O\left( d_0 \exp\left( -\frac{d_0}{2} \log\left( \frac{d_0}{\log d_0} \right) \right) \right) \right) \right.$$

$$\left. \cdot \left( 1 - \sqrt{\frac{8}{\pi}} \frac{1}{d_1^* d_0^{1/2}} + O\left( \frac{1}{d_1^* d_0^{3/2}} + \frac{1}{d_1^{*2} d_0 N} + d_1^* 2^{-N} + d_1^* d_0 \exp\left( -\frac{d_0}{2} \log\left( \frac{d_0}{\log d_0} \right) \right) \right) \right) \right)$$

$$= \sqrt{\frac{8}{\pi}} \frac{1}{d_0^{1/2}} + \frac{2 d_0^{1/2} \sqrt{\log d_0}}{N} + O\left( \frac{1}{d_0^{3/2}} + \frac{d_0^{1/4}}{d_1^* N} + d_1^* 2^{-N} + d_1^* d_0 \exp\left( -\frac{d_0}{2} \log\left( \frac{d_0}{\log d_0} \right) \right) \right).$$

Thus, taking the log, and using $\log(1 - x) = -x + O(x^2)$, we obtain, for $\sin \alpha = 1/(d_1^* d_0 N)$

$\log \mathbb{P}(\mathbf{X} \in \mathcal{M}^\alpha(\mathbf{W}^*))$

$$\geq \log \left( 1 - \sqrt{\frac{8}{\pi}} \frac{1}{d_0^{1/2}} - \frac{2 d_0^{1/2} \sqrt{\log d_0}}{N} + O\left( \frac{1}{d_0^{3/2}} + \frac{d_0^{1/4}}{d_1^* N} + d_1^* 2^{-N} + d_0 \exp\left( -\frac{d_0}{2} \log\left( \frac{d_0}{\log d_0} \right) \right) \right) \right)$$

$$= -\sqrt{\frac{8}{\pi}} \frac{1}{d_0^{1/2}} - \frac{2 d_0^{1/2} \sqrt{\log d_0}}{N} + O\left( \frac{1}{d_0^{3/2}} + \frac{d_0^{1/4}}{d_1^* N} + d_1^* 2^{-N} + d_0 \exp\left( -\frac{d_0}{2} \log\left( \frac{d_0}{\log d_0} \right) \right) \right).$$

Recall that $d_1^* \triangleq 4 \lceil N/(2d_0 - 2) \rceil \doteq N/d_0$. Taking the limit $N \to \infty$, $d_0 \to \infty$ with $d_1^* \dot{\leq} d_0 \dot{\leq} N$, we have

$$\mathbb{P}(\mathbf{X} \notin \mathcal{M}^\alpha(\mathbf{W}^*)) \dot{\leq} 1 - \exp\left( -\sqrt{\frac{8}{\pi}} d_0^{-1/2} - \frac{2 d_0^{1/2} \sqrt{\log d_0}}{N} \right) \leq \sqrt{\frac{8}{\pi}} d_0^{-1/2} + \frac{2 d_0^{1/2} \sqrt{\log d_0}}{N}$$

$\square$

# Part III

# Numerical Experiments - implementation details

Code and trained models for CIFAR and ImageNet results is available here `https://github.com/MNNsMinima/Paper`. In MNIST, CIFAR and ImageNet we performed binary classification on between the original odd and even class numbers. In we performed this binary classification between digits $0 - 4$ and $5 - 9$. Weights were initialized to be uniform with mean zero and variance $2/d$, where $d$ is fan-in (here the width of the previous neuron layer), as suggested in (He et al., 2015). In each epoch we randomly permuted the dataset and used the Adam (Kingma & Ba, 2014) optimization method (a variant of SGD) with $\beta_1 = 0.9, \beta_2 = 0.99, \varepsilon = 10^{-8}$. Different learning rates and mini-batch sizes were selected for each dataset and architecture. In CIFAR10 and ImageNet we used a learning-rate of $\alpha = 10^{-3}$ and a mini-batch size of 1024; also, ZCA whitening of the training samples was done to remove correlations between the input dimensions, allowing faster convergence. We define $L$ as the number of weight layers. For the random dataset we use a mini-batch size of $\lfloor \min(N/2, d/2) \rfloor$ with learning rate $\alpha = 0.1$ and 0.05, for $L = 2$ and 3, respectively. In the random data parameter scans the training was done for no more than 4000 epochs – we stopped if $\text{MCE} = 0$ was reached.

