# OpenReview forum: "Exponentially vanishing sub-optimal local minima in multilayer neural networks"
_ICLR.cc/2018/Conference — Invite to Workshop Track_

### Official Review · AnonReviewer1 · 2017-11-23
**Interesting result despite clear limitations**

**Rating:** 7
**Confidence:** 2

**Review:**

This paper studies the question: Why does SGD on deep network is often successful, despite the fact that the objective induces bad local minima?
The approach in this paper is to study a standard MNN with one hidden layer. They show that in an overparametrized regime, where the number of parameters is logarithmically larger than the number of parameters in the input, the ratio between the number of (bad) local minima to the number of global minima decays exponentially. They show this for a piecewise linear activation function, and input drawn from a standard Normal distribution. Their improvement over previous work is that the required overparameterization is fairly moderate, and that the network that they considered is similar to ones used in practice.

This result seems interesting, although it is clearly not sufficient to explain even the success on the setting studied in this paper, since the number of minima of a certain type does not correspond to the probability of the SGD ending in one: to estimate the latter, the size of each basin of attraction should be taken into account. The authors are aware of this point and mention it as a disadvantage. However, since this question in general is a difficult one, any progress might be considered interesting. Hopefully, in future work it would be possible to also bound the probability of starting in one of the basins of attraction of bad local minima.

The paper is well written and well presented, and the limitations of the approach, as well as its advantages over previous work, are clearly explained. As I am not an expert on the previous works in this field, my judgment relies mostly on this work and its representation of previous work. I did not verify the proofs in the appendix.

---

> ### Author Response · Authors · 2017-12-20
> **Please read our main response on the submission forum**
>
> We thank the reviewer for his positive review. We hope our main response in the submission forum clarified some of the uncertainty regarding our novelty.

---

### Official Review · AnonReviewer3 · 2017-11-27
**Assumptions are more realistic, but the main results are less motivated.**

**Rating:** 6
**Confidence:** 3

**Review:**

## Summary
This paper aims to tackle the question: "why does standard SGD based algorithms on neural network converge to 'good' solutions?"

Pros:
Authors ask the question of convergence of optimization (ignoring generalization error): how "likely" is that an over-parameterized (d1d0 > N) single hidden layer binary classifier "find" a good (possibly over-fitted) local minimum. They make a set of assumptions (A1-A3) which are weaker (d1 > N^{1/2}) than the ones used earlier works. Previous works needed a wide hidden layer (d1 > N).

Assumptions (d0=input dim, d1=hidden dim, N=n of datapoints, X=datapoints matrix):
A1. Datapoints X come from a Gaussian distribution
A2. N^{1/2} < d0 =< N
A3. N polylog(N) < d0d1 (approximate n of. parameters)  and d1 =< N

This paper proves that total "angular volume" of "regions" (defined with respect to the piecewise linear regions of neuron activations) with differentiable bad-local minima are exponentially small when compared with to the total "angular volume" of "regions" containing only differentiable global-minimal. The proof boils down to counting arguments and concentration inequality.

Cons:
Non-differentiable stationary points are left as a challenging future work on this paper. Non-differentiability aside, authors show a possible way by which shallow neural networks might be over-fitting the data. But this is only half the story and does not completely answer the question. First, exponentially vanishing (in N) volume of the "regions" containing bad-local minima doesn't mean that the number of bad local minima are exponentially small when compared to number global minima.
Secondly, as the authors aptly pointed out in the discussion section, this results doesn't mean neural networks will converge to good local minima because these bad local minimas can have a large basins of attraction.
Lastly, appropriate comparisons with the existing literature is lacking. It is hinted that this paper is more general as the assumptions are more realistic. However, it comes at a cost of losing sharpness in the theoretical results. It is not well motivated why one should study the angular volume of the global and local minima.

## Questions and comments
1. How critical is Gaussian-datapoints assumption (A1)? Which part of the proof fails to generalize?
2. Can the proof be extended to scalar regression?  It seems hard to generalize to vector output neural networks. What about deep neural networks?
3. Can you relate the results to other more recent works like: https://arxiv.org/pdf/1707.04926.pdf.
4. Piecewise linear and positively homogeneous (https://arxiv.org/pdf/1506.07540.pdf) activation seem to be important assumption of the paper. It should probably be mentioned explicitly.
5. In the experiments section, it is mentioned that "...inputs to the hidden neurons converge to a distinctly non-zero value. This indicates we converged to DLMs." How can you guarantee that it is a local minimum and not a saddle point?

---

> ### Author Response · Authors · 2017-12-20
> **Please read our main response on the submission forum**
>
> ## Reply to general comments
>
> [“Exponentially vanishing (in N) volume of the "regions" containing bad-local minima doesn't mean that the number of bad local minima are exponentially small when compared to number global minima.”, “It is not well motivated why one should study the angular volume of the global and local minima.”]
>
> A explained in section 3, the “number” of local minima is not a well-defined, in the over-parameterized regime. In this case local minima are not points, but linear manifolds (e.g., lines, hyperplanes) within each differentiable region, since there are certain directions in which we can change the weights and do not modify the loss. Instead, one can try to count the numbers of “local minima manifolds” of each type (which are equal the number of differentiable regions containing bad/good minima). However, this can be misleading since some minima occupy much larger regions than others. To take this into account we therefore chose to bound the total (angular) volume of the regions for each type (the regular volume is infinite). Incidentally, we also bound the number of regions (equal to the number of “local minima manifolds”) in the derivation of the total volume bounds, since we use a product of the two worst case bounds on (number of regions)*(single region volume). This is the reason we focused on the “angular volume” of local minima, as the strongest possible interpretation we could think of the “number” of local minima. We can clarify this further in the paper, if needed.
>
> [“Appropriate comparisons with the existing literature is lacking.” “It is hinted that this paper is more general as the assumptions are more realistic. However, it comes at a cost of losing sharpness in the theoretical results.” “As the authors aptly pointed out in the discussion section, this results doesn't mean neural networks will converge to good local minima because these bad local minimas can have a large basins of attraction.”]
>
> Please see our main response in the submission forum. We believe that advancing theory on realistic models is more important then proving strong claims on highly non-realistic models. In other words, though other papers prove seemingly stronger results, this was always at the high price of being unrealistic and therefore very far from practical usage, as we review in the introduction and the our main response in the submission forum.
>
> ## Reply to specific questions and comments
>
> 1) The Gaussian assumption could be relaxed to other near-isotropic distributions (e.g., sparse-land model, (Elad, 2010, Section 9.2)), as scale constants do not affect any of the calculations. If the input is non-isotropic then it could harm several probabilistic proofs: First, the bound on P(WX>0) in Lemma 16 could be much worse, which can harm the proof of theorem 6 (upper bound on sub-optimal local minima). Second, the bound on probability for a certain angular margin (Lemma 22 and 23) could also become worse, which will harm the proof of Theorem 9 (lower bound on global minima).
>
> 2) Extension to scalar regression mainly requires the extension of the Theorem 8 to this case. Which we believe is quite possible, yet outside the scope of this paper. Our results apply also to multilayer neural network with more then single hidden layer if only the last two layers are trained, and our assumptions hold with respect to those two layers, as we discuss in the introduction. Therefore, it suggests that reaching zero training error might be easy even in more complicated neural nets with over-parameterization in the two last layers (e.g., Alexnet). We believe that extending our results to deep networks where all the layers are optimized, and to multi-output case, is challenging, yet possible, and requires much more work, as we mention in the discussion.
>
> 3) Please see our main response in the submission forum.
>
> 4) Please see our response on the Haeffele & Vidal paper in main submission forum. We state explicitly both in the abstract and introduction that we focus on neural nets “with one hidden layer of piecewise linear units” (this is in the first line to our discussion of the results in both cases). Since essentially all piecewise linear units used in practice (e.g., ReLU) are “positively homogeneous”, we did not mention this explicitly.
>
> 5) Good point. It is easy to show that saddle points in a single hidden layer network must have zero weights in the last layer, and we can verify numerically this is not the case. However, the main point in this paragraph was to show we do not converge to a non-differentiable critical point, so we simply changed the phrasing in the last sentence to “This indicates we did not converge to converged to non-differentiable critical points."

---

### Official Review · AnonReviewer2 · 2017-11-27
**Not very surprising. Marginal insight on top of prior literature.**

**Rating:** 5
**Confidence:** 3

**Review:**

This is a theory paper. The authors consider networks with single hidden layer. They assume gaussian input and binary labels. Compared to some of the existing literature, they study a more realistic model that allows for mild overparametrization and approximately speaking d_0=d_1=sqrt(N). The main result is that volume of suboptimal local minima exponentially decreases in comparison to global minima.

In my opinion, paper has multiple drawbacks.
1) Lack of surprise factor: There are already multiple papers essentially saying similar things. I am not sure if this contributes substantially on top of existing literature.
2) Lack of algorithmic results: While the volume of suboptimal DLM being small is an interesting result, it doesn't provide substantial algorithmic insight. Recent literature contains results that states not only all locals are global but also gradient descent provably converges to the global with a good rate. See Soltanolkotabi et al.
3) Mean squared error for classification problem (discrete labels) does not sound reasonable to me. I believe there are already some zero error results for continuous labels. Logistic loss would have made a more compelling story.

Minor comments:
i) Results are limited to single hidden layer whereas the title states multilayer. While single hidden layer is multilayer, stating single hidden layer upfront might be more informative for the reader.
ii) Theorem 10 and Theorem 6 essentially has the same bound on the right hand side but Theorem 10 additionally divides local volume by global which decreases by exp(-2Nlog N). So it appears to me that Thm 10 is missing an additional exp(2Nlog N) factor on the right hand side.

Revision (response to authors): I appreciate the authors' response and clarification. I do agree that my comparison to Soltanolkotabi missed the fact that his result only applies to quadratic activations for global convergence (also many thanks to Jason for clarification). Additionally, this paper appeared earlier on arXiv. In this sense, this paper has novel technical contribution compared to prior literature. On the other hand, I still think the main message is mostly covered by existing works. I do agree that squared-loss can be used for classification but it makes the setup less realistic. Finally, while introduction discusses the "last two layers", I don't see a technical result proving that the results extends to the last two layers of a deeper network. At least one of the assumptions require Gaussian data and the input to the last two layers will not be Gaussian even if all previous layers are fixed. Consequently, the "multilayer" title is somewhat misleading.

---

> ### Author Response · Authors · 2017-12-20
> **Please read our main response on the submission forum**
>
> ## Reply to general comments
>
> [“Results of similar flavor already exists”, “There are already multiple papers essentially saying similar things”, “ I believe there are already some zero error results for continuous labels.”, “Recent literature contains results that states not only all locals are global but also gradient descent provably converges to the global with a good rate. See Soltanolkotabi et al.“]
>
> We believe there has been a misunderstanding: no “vanishing bad local minima”/“zero error”/”convergence to global minimum” results have been proven without using highly unrealistic assumptions, as we clarify in our or main response (detailed in the submission forum). If we understood correctly, all major concerns of the reviewer stem from this misunderstanding. We hope we clarified this issue.
>
> ## Reply to Minor comments:
>
> [ Theorem 10 and Theorem 6 essentially has the same bound on the right hand side but Theorem 10 additionally divides local volume by global which decreases by exp(-2Nlog N). So it appears to me that Thm 10 is missing an additional exp(2Nlog N) factor on the right hand side.]
>
> This is not an error. There are two bounds on the global minima volume in Theorem 9. To prove theorem 10, we use the left bound (exp(-d_1^*d_0 logN) which is better than the right bound exp(-2Nlog N). Specifically, this left bound becomes negligible in comparison to the bound of Theorem 6 (from assumption 4) so it has no effect on the final bound of Theorem 10.
>
> [ Logistic loss would have made a more compelling story.]
>
> Yes, logistic loss is indeed better for classification. However, note that (1) almost all previous theory paper use quadratic error, (2) yet, to the best of our knowledge, as we clarified in our main response, there are no zero error results for continuous labels with realistic assumptions. (3) It is possible to do binary classification also with quadratic loss. In this paper we aimed to find the simplest case where the property of vanishing “bad” local minima could be proved for the first time under reasonably realistic conditions. We believe it will not be very hard to extend our results to logistic loss, as we write in the discussion, but this analysis is outside the scope of this paper.
>
> [Results are limited to single hidden layer whereas the title states multilayer. While single hidden layer is multilayer, stating single hidden layer upfront might be more informative for the reader.]
>
> We can make this modification if the reviewer insists, but as this information is already written in the abstract, we believe that changing “multilayer” to “single hidden layer” will make the title a bit too long (also, using “two-layer” instead is a bit vague, as some people call such a network “three-layer”). Furthermore, it will somewhat undersell this paper, as our results relate also to multilayer neural network with more then single hidden layer if only the last two layers are trained, and our assumptions hold with respect to those two layers, as we discuss in the introduction. This suggests that reaching zero training error might be easy even in more complicated neural nets with over-parameterization in the two last layers (e.g., Alexnet).

---

> ### Author Response · Authors · 2018-01-12
> **Response to reviewer revision**
>
> After our revision, the only remaining major concern of the reviewer is that
> "the main message is mostly covered by existing works."
>
> However, it is not clear to us what existing work the reviewer is referring to. If the reviewer is saying that similar results were proved earlier, then we disagree. In any case, would like to know which results the reviewer is referring to (even so that we can revise the paper to address any such concerns).
>
> If the reviewer is saying instead that our results on "vanishing bad local minima" are not surprising because many previous papers already conjectured this (or proved this under unrealistic conditions), then we believe that:
> (1) There is some value in advancing towards rigorous proofs of "unsurprising" important conjectures (e.g., P != NP).
> (2) This issue is not well understood as the reviewer suggests, since, under similar conditions, optimization with respect to the expected loss (instead of the empirical loss, which we used here) can converge to local minima: https://arxiv.org/abs/1712.08968. This suggests we still do not understand the complexity of this issue, even in the most basic settings.

---

### Author Response · Authors · 2017-12-20
**Main response to reviewers comments**

We sincerely thank the reviewers for their feedback on our paper. We believe that the major concerns may have been a result of a misunderstanding of previous literature. Specifically, in several previous papers the results may appear stronger then they truly are, if one misses an unrealistic assumption buried in the mathematical details, as we explain below, for all the papers mentioned by the reviewers.

First, the reviewers mention Soltanolkotabi et al. (https://arxiv.org/pdf/1707.04926.pdf., which we already cite in the paper, and originally appeared after us) as a previous paper that proved stronger results. However, these results require highly unrealistic assumptions regarding the initialization or activation functions. Specifically, as we mentioned in our paper, the main result in this paper (Theorem 2.5) unrealistically assumes that the weights are initialized very close to the target weights of the teacher generating the labels: see eq. 2.4 in this paper, and recall that k (# neurons)>=d (input dimension) << n (number of samples), so this distance is ~ (d/n)^(1.5)), which is typically very small. Other theorems in this paper assume quadratic activation functions, which is also unrealistic (e.g., such network can only approximate quadratic functions). We confirmed this in a personal communication with an author of Soltanolkotabi et al., who also agreed the assumptions in our paper are significantly more realistic then this paper, and also in comparison to other related results. Lastly, in case the reviewers had in mind another paper by Soltanolkotabi (https://arxiv.org/abs/1705.04591), it only examined the case of a *single* ReLU neuron.

Second, in the paper by Haeffele & Vidal (https://arxiv.org/pdf/1506.07540.pdf) the main result requires unrealistically wide neural layers: the condition r>card(D) in Theorem 17, when applied to a neural net with single hidden layer, implies that the number of neurons is larger then (input dimension)*(number of samples). For such extremely large layers, it is easy to get zero error by optimizing the last (linear) layer alone (where #variables > #samples), like we discuss in the introduction in the case of extremely wide layers (d_{L-1}>N). We now cite it together with the list of other works that also assumed extremely wide layers (this list was not meant to be exhaustive).

We hope our answers below will help clarify any such misunderstandings. To emphasize, no previous paper rigorously proved similar results without requiring highly unrealistic assumptions (either a heavily modified neural net model or training method, strong assumptions on the labels (e.g., “near” linear separability), or an unrealistically wide hidden layer with more units then data samples). This prevents these previous results from being used in practice, and indicates the inherent difficulty of such proofs. In contrast to previous works, the results in our paper are applicable in *some* situations (e.g., Gaussian data) where a neural net trained using SGD might be used and be useful (e.g., have a better performance then a linear classifier). Therefore, we feel that our results are a step in the direction of a global convergence proof, for a reasonably realistic models. We feel that trying to close the gap towards a “convergence to global minimum” proof for such models is a worthy goal, given that such a proof seems far from reach, despite many years of research and many papers on the subject. In the discussion we suggest how to extend our results towards this final goal.

Revision summary: Added references by Haeffele & Vidal, and clarified a sentence in the experimental section, following a a comment by reviewer 3.

Additional comments are answered individually for each reviewer.

---

### Decision · Program_Chairs · 2018-01-29
**ICLR 2018 Conference Acceptance Decision**

**Decision:**

Invite to Workshop Track

**Comment:**

The paper analyzes neural network with hidden layer of piecewise linear units, a single output, and a quadratic loss. The reviewers find the results incremental and not "surprising", and also complained about comparison with previous work. I think the topic is very pertinent, and definitely more relevant compared to studying multi-layer linear networks. Hence, I recommend the paper be presented in the workshop track.